# Adaptive Sigmoid Clipping for Balancing the Direction–Magnitude Mismatch Trade-off in Differentially Private Learning

**Faeze Moradi Kalarde**
ECE Department
University of Toronto
faeze.moradi@mail.utoronto.ca

**Ali Bereyhi**
ECE Department
University of Toronto
ali.bereyhi@utoronto.ca

**Ben Liang**
ECE Department
University of Toronto
liang@ece.utoronto.ca

**Min Dong**
ECSE Department
Ontario Tech University
Min.Dong@ontariotechu.ca

## Abstract

Differential privacy (DP) limits the impact of individual training data samples by bounding their gradient norms through clipping. Conventional clipping operations assign unequal scaling factors to sample gradients with different norms, leading to a direction mismatch between the true batch gradient and the aggregation of the clipped gradients. Applying a smaller but identical scaling factor to all sample gradients alleviates this direction mismatch; however, it intensifies the magnitude mismatch by excessively reducing the aggregation norm. This work proposes a novel clipping method, termed adaptive sigmoid (AdaSig), which uses a sigmoid function with an adjustable saturation slope to clip the sample gradients. The slope is adaptively adjusted during the training process to balance the trade-off between direction mismatch and magnitude mismatch, as the statistics of sample gradients evolve over the training iterations. Despite AdaSig's adaptive nature, our convergence analysis demonstrates that differentially private stochastic gradient descent (DP-SGD) with AdaSig clipping retains the best-known convergence rate under non-convex loss functions. Evaluating AdaSig on sentence and image classification tasks across different datasets shows that it consistently improves learning performance compared with established clipping methods.

## 1 Introduction

Deep learning models can expose individual data samples to privacy risks and are vulnerable to multiple types of practical attacks [2, 5, 26, 29]. To mitigate these risks, differential privacy (DP) [10] has emerged as a widely adopted standard in machine learning. Differentially private training constrains the impact of individual data samples on the training process, ensuring that the resulting model statistically resembles the one trained by the exclusion of an arbitrary data sample. As a result, inferring the presence of an individual data sample in the training set becomes challenging when observing the model during training. Differentially private stochastic gradient descent (DP-SGD) is a prominent privacy-preserving algorithm that incorporates DP into the SGD algorithm through two steps: (i) bounding sensitivity through clipping, and (ii) introducing uncertainty by adding Gaussian noise [1]. The first step limits the $\ell_2$-sensitivity of the batch gradient against the addition or removal of data samples by upper bounding the magnitude of the sample gradients using a clipping operation.

A widely used clipping operation, which we refer to as *vanilla clipping* [1, 8], scales each sample gradient $\mathbf{g}_{i,t}$ with a norm greater than $C$ to constrain its norm to $C$, as

$$\tilde{\mathbf{g}}_{i,t} = \mathbf{g}_{i,t} \min\left(1, \frac{C}{\|\mathbf{g}_{i,t}\|}\right), \tag{1}$$

where the constant $C$ is termed the *clipping threshold*. In the second step, a zero-mean Gaussian noise is added to the aggregation of the clipped gradients with a standard deviation that is proportional to the clipping threshold.

Applying the clipping operation to sample gradients introduces a deviation between the aggregation of clipped gradients, i.e., $\sum_{i \in \mathcal{B}_t} \tilde{\mathbf{g}}_{i,t}$, and the true batch gradient, i.e., $\sum_{i \in \mathcal{B}_t} \mathbf{g}_{i,t}$, which is referred to as *bias* [3, 13, 19]. The bias caused by clipping manifests in two forms: (i) *direction deviation*, and (ii) *magnitude deviation* from the true batch gradient. Direction deviation can be particularly severe in many classical clipping methods that are based on vanilla clipping operation in (1). This follows from the fact that the vanilla clipping maps all gradient norms greater than $C$ to $C$, and hence neglects the information contained in the diversity of the sample gradient norms within the batch. This can be interpreted as assigning unequal weights to sample gradients, which causes a significant direction deviation. This misalignment can steer training steps towards undesired regions of the loss landscape and hinder training convergence [7, 11].

In response, the per-sample adaptive clipping (PSAC) method [36] was proposed to mitigate the direction deviation. Figure 1 shows the variation of the clipped gradient norm $\|\tilde{\mathbf{g}}_{i,t}\|$ versus the gradient norm $\|\mathbf{g}_{i,t}\|$ for the vanilla clipping and compares it with PSAC for the same threshold $C = 0.1$. We also include the curve for the automatic clipping method (Auto-S) [4], which is proposed to eliminate the need for tuning the clipping threshold. This figure shows the effectiveness of each method in reducing direction deviation: the more closely a curve can be approximated by a line passing through the origin, the better the clipping method approximates equal scaling of gradient samples with different magnitudes. We observe that the vanilla clipping method hard-clips the magnitude of sample gradients that are larger than $C$, resulting in a limited linear region[1] on the curve. Using Auto-S, the hard-thresholding behavior of the vanilla method is slightly reduced;

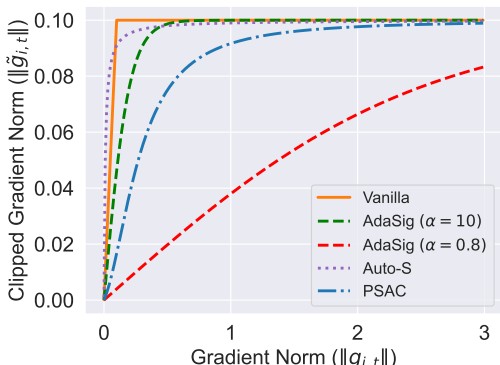

Figure 1: Clipped gradient norm vs. gradient norm. Comparing linear regions for different clipping methods. PSAC has a wider linear region compared with the vanilla clipping. The linear region of AdaSig can be adjusted by altering $\alpha$.

nonetheless, its restricted linear region leads to a similar scaling as the vanilla method. In contrast, the PSAC clipping method has a wider linear region than vanilla clipping and Auto-S, as it preserves the same scaling factor across a broader range of sample gradient norms, thereby providing a more accurate approximation of equal scaling.

While expanding the linear region can reduce direction deviation, it also decreases the norm of the aggregation of the clipped gradients compared with that of the true batch gradient (i.e., larger magnitude deviation). This occurs because the wide linear region of the clipping curve results in a small scaling factor, causing sample gradients with smaller magnitudes to have excessively small norms after scaling. This suggests that the span of the linear region on the clipping curve needs to be carefully adjusted to better balance the trade-off between direction deviation and magnitude deviation. Moreover, since the sample gradient statistics change across different training iterations, it is necessary to adaptively adjust the linear region to balance the trade-off throughout the training process.

**Contributions** Motivated by the above observation, we propose a novel clipping method called Adaptive Sigmoid (AdaSig), where the magnitude of the clipped gradient is determined from the original gradient magnitude using a scaled and shifted sigmoid function. AdaSig features a saturation slope parameter that adaptively adjusts the span of the linear region throughout training to balance the trade-off between direction and magnitude mismatches.

---

[1]"Linear region" is used in contrast to the "lazy region" to refer to the part of the curve that can be well approximated by a straight line passing through the origin.

Figure 1 also shows the clipping curves for two choices of the saturation slope (denoted by $\alpha$) in AdaSig. As $\alpha$ shrinks, the linear region of the clipping operation expands resulting in better approximation of equal gradient scaling and thus smaller *direction deviation*. Conversely, larger $\alpha$ preserves the magnitude of smaller sample gradients thus reducing the *magnitude deviation*. To adaptively adjust the clipping operation based on the gradient statistics throughout training, we treat $\alpha$ as a learnable parameter and update it in each iteration to minimize the empirical loss. To the best of our knowledge, this is the first study that designs a clipping method to balance the trade-off between the magnitude deviation and direction deviation. This work makes the following contributions:

- We introduce AdaSig, a novel clipping strategy that allows balancing the direction deviation and magnitude deviation by adjusting a parameter $\alpha$. We treat $\alpha$ as a learnable parameter and derive its SGD-based update for empirical loss minimization.

- We develop a new algorithm for DP-SGD with AdaSig clipping (DP-SGD-AdaSig), by establishing an update rule for $\alpha$ that effectively preserves privacy.

- AdaSig's unique clipping structure and varying slope complicate its convergence analysis such that existing analyses are inapplicable. However, we derive a convergence bound for DP-SGD-AdaSig in non-convex loss settings, showing that its privacy–utility trade-off matches the best-known bound in the literature.

- We conduct experiments on image and sentence classification tasks. Performance comparison with existing clipping methods shows the efficacy of AdaSig in learning enhancement through a proper balance between direction deviation and magnitude deviation.

## 2    Related Work

**Extensions of Vanilla Clipping** In vanilla clipping, the clipping threshold substantially impacts training performance [19]. Large $C$ results in large privacy noise variance, while small $C$ leads to aggressive gradient clipping, causing a significant bias. Several studies adaptively adjust $C$ to improve the bias–variance trade-off. In [3], the optimal threshold is estimated using gradient quantiles during training. In [13], the clipping threshold is treated as a learnable parameter, allowing it to be optimized dynamically over the training iterations. However, these works focus on the bias–variance trade-off for vanilla clipping and cannot balance the magnitude and direction deviations introduced by the bias for a *fixed* variance level. This limitation arises from the inherent nature of vanilla clipping, where reducing the direction deviation (increasing $C$) inevitably increases the variance.

Some studies combine vanilla clipping with error feedback to mitigate the bias introduced by clipping [41, 16]. This technique is applicable to any clipping method, including ours, and is orthogonal to the focus of this study.

**Other Clipping Methods** Automatic clipping (Auto-S) [4] and normalized SGD (NSGD) [39] are similar approaches that aim to eliminate the need for tuning the clipping threshold. Their key idea is to normalize gradients as $\tilde{\mathbf{g}}_{i,t} = \frac{C\mathbf{g}_{i,t}}{\|\mathbf{g}_{i,t}\|+r}$ for some small positive constant $r$. Although these methods are effective for their main goal, they suffer considerably from direction deviation, as they assign larger weights to gradients with smaller magnitudes. These small gradients are often opposite to the true batch gradient direction, as empirically shown in [36]. To address the direction deviation issue, PSAC [36] modifies the clipping operation to $\tilde{\mathbf{g}}_{i,t} = \frac{C\mathbf{g}_{i,t}}{\|\mathbf{g}_{i,t}\| + \frac{r}{\|\mathbf{g}_{i,t}\|+r}}$ for a positive $r$. This adjustment reduces the weights assigned to small gradients and thus mitigates the direction deviation. While PSAC effectively reduces the direction deviation, it cannot adaptively balance the trade-off between magnitude and direction deviations during training because its linear region span remains fixed throughout the process (Figure 1). To address this limitation, we propose AdaSig, which dynamically adjusts its saturation slope to better balance this trade-off.

## 3    Background and Preliminaries

**Learning Model and SGD** We consider a supervised setting, where the training dataset consists of $N$ samples, denoted as $\mathcal{D} = \{(x_i, y_i)\}_{i=1}^N$, with $x_i \in \mathcal{X}$ representing the vector of input features and $y_i \in \mathcal{Y}$ denoting the corresponding target output. The goal is to learn a model $f_{\boldsymbol{\theta}} : \mathcal{X} \to \mathcal{Y}$, with parameters $\boldsymbol{\theta} \in \mathbb{R}^d$, that maps input features to the target outputs. The discrepancy between

the predicted outputs $\hat{y}_i = f_{\boldsymbol{\theta}}(x_i)$ and the actual targets $y_i$ is measured by the sample loss function, denoted as $\ell : \mathcal{Y} \times \mathcal{Y} \to \mathbb{R}$. Let $h^i(\boldsymbol{\theta})$ denote the loss of sample $i$ as a function of the model parameters, i.e., $h^i(\boldsymbol{\theta}) \triangleq \ell(f_{\boldsymbol{\theta}}(x_i), y_i)$. The population loss is defined as $L(\boldsymbol{\theta}) = \mathbb{E}_{(x,y) \sim \mathcal{P}}\big[\ell(f_{\boldsymbol{\theta}}(x), y)\big]$, where $\mathcal{P}$ denotes the unknown data distribution. In iteration $t$, stochastic gradient descent (SGD) updates the model parameters using the average gradient of the loss over the samples within a batch of data as

$$\boldsymbol{\theta}_{t+1} = \boldsymbol{\theta}_t - \frac{\lambda}{B} \sum_{i \in \mathcal{B}_t} \nabla_{\boldsymbol{\theta}} h^i(\boldsymbol{\theta})\Big|_{\boldsymbol{\theta} = \boldsymbol{\theta}_t}, \tag{2}$$

where $\nabla_{\boldsymbol{\theta}} h^i(\boldsymbol{\theta})$ denotes the gradient of the loss function with respect to (w.r.t.) the model parameters, $\boldsymbol{\theta}_t$ denotes the model at iteration $t$, $\mathcal{B}_t \subseteq \mathcal{D}$ is the batch of data at iteration $t$, $B$ denotes the expected batch size, and $\lambda$ is the learning rate.

**Differential Privacy** The randomized mechanism $M : \mathcal{S} \to \mathcal{R}$ with domain $\mathcal{S}$ and range $\mathcal{R}$ satisfies $(\epsilon, \delta)$-differential privacy (DP) if, for any two *neighboring* datasets $S, S' \in \mathcal{S}$, i.e., $S'$ is formed by adding or removing a single sample from $S$, and for any output set $R \subseteq \mathcal{R}$,

$$\Pr[M(S) \in R] \le e^{\epsilon} \Pr[M(S') \in R] + \delta, \tag{3}$$

where $\Pr[A]$ denotes the probability of the event $A$. Inequality (3) implies that for small $\epsilon$ and $\delta$, the output distributions of the mechanism under neighboring datasets approach each other, making it harder to detect the contribution of an individual sample from the mechanism's output.

**Differentially Private SGD (DP-SGD)** Let $\mathbf{g}_{i,t} \in \mathbb{R}^d$ denote the gradient of loss for sample $i$ at iteration $t$, i.e., $\mathbf{g}_{i,t} \triangleq \nabla_{\boldsymbol{\theta}} h^i(\boldsymbol{\theta})|_{\boldsymbol{\theta} = \boldsymbol{\theta}_t}$. DP-SGD achieves privacy protection via two steps: (i) The gradient samples within the batch are first passed through a clipping method that returns a clipped gradient $\tilde{\mathbf{g}}_{i,t}$ whose $\ell_2$-norm is bounded by a constant $C$. Basic DP-SGD uses the vanilla clipping method in (1). The aim of clipping is to bound the $\ell_2$-sensitivity of the aggregation of the gradients by $C$. (ii) After aggregating the clipped gradients over the batch, zero-mean Gaussian privacy noise is added to introduce uncertainty. To ensure the privacy of training, the variance of the noise process is scaled with $C^2$. Consequently, in iteration $t$ of DP-SGD, we update the model parameters as

$$\boldsymbol{\theta}_{t+1} = \boldsymbol{\theta}_t - \frac{\lambda}{B} \Big( \sum_{i \in \mathcal{B}_t} \tilde{\mathbf{g}}_{i,t} + \mathbf{n}_t \Big), \tag{4}$$

where $\mathbf{n}_t \sim \mathcal{N}(0, C^2 \sigma^2 \mathbf{I}_d)$ is the Gaussian privacy noise with $\sigma$ being the noise multiplier determined by the privacy budget $(\epsilon, \delta)$.

## 4  AdaSig Clipping

Vanilla clipping in (1) outputs a vector with the smallest error norm relative to the input vector. It however introduces a notable *direction deviation* between the true batch gradient and the sum of clipped gradients due to unequal scaling of different sample gradients. A naive solution to this issue is to scale all sample gradients by the same coefficient such that all resulting norms fall below the clipping threshold. This approach requires the scaling coefficient to be small enough so that the largest gradient norm is reduced to the clipping threshold. As a result, gradients with smaller norms are scaled down to very small norms, leading to an aggregation with excessively small magnitude (*magnitude deviation*). Since the privacy noise is proportional to the clipping threshold, this low-magnitude aggregation results in a very low signal-to-noise ratio, which hinders training performance.

In light of the above discussion, a proper clipping operation should not only have a bounded range to constrain the $\ell_2$ sensitivity, but it should also ensure that the output norm is monotonically increasing w.r.t. the input norm, with an adjustable linear region span that balances the trade-off between direction and magnitude deviations. Since the sigmoid function, with an adjustable saturation slope, satisfies both of these properties, we propose a clipping operation based on the sigmoid function, termed adaptive sigmoid (AdaSig). Further details on the choice of the sigmoid function for clipping are provided in Appendix A. This operation is illustrated in the sequel.

**Clipping with Sigmoid Function** For the clipping threshold $C$, we define AdaSig clipping as

$$\psi_{\alpha}(x) = C \Big( \frac{2}{1 + e^{-\alpha x}} - 1 \Big), \tag{5}$$

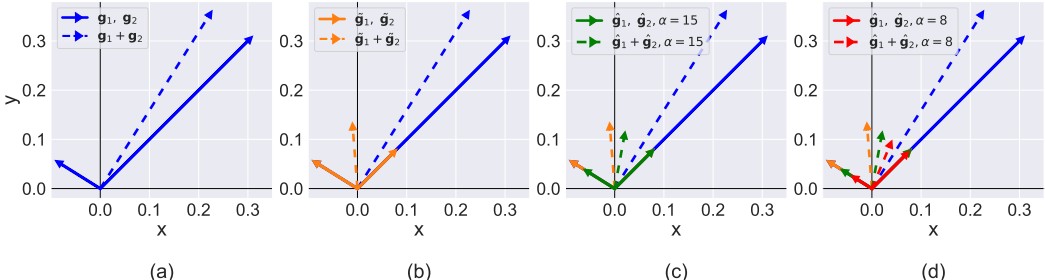

Figure 2: Illustrative example: sample gradients are shown by solid arrows and their aggregation is indicated by a dashed arrow. The horizontal and vertical axes correspond to the first and second entries of the two-dimensional vectors, respectively. (a) shows true gradients and their summation. (b) shows the gradients clipped by the vanilla method and their aggregation in orange. (c) and (d) show the gradients clipped by AdaSig for $\alpha = 15$ and $\alpha = 8$ in green and red, respectively. The adjustability of $\alpha$ enables AdaSig to balance the trade-off between direction and magnitude deviations.

where $\alpha > 0$ is the saturation slope. Using the AdaSig function, the sample gradient $\mathbf{g}_{i,t}$ is clipped as

$$\tilde{\mathbf{g}}_{i,t} = \psi_\alpha\big(\|\mathbf{g}_{i,t}\|\big) \frac{\mathbf{g}_{i,t}}{\|\mathbf{g}_{i,t}\|}. \tag{6}$$

One can see that this clipping addresses the two mentioned desired properties: (i) since $\psi_\alpha(x) \in [0, C)$ for $x \geq 0$, the sum of the clipped gradients by AdaSig has a bounded $\ell_2$-sensitivity equal to $C$. (ii) The scaling is a monotonic function of the input norm and can adjust its linear region by varying $\alpha$. Figure 1 shows the clipping curves of AdaSig for two choices of $\alpha$. As the figure shows, the parameter $\alpha$ enables us to sketch an adjustable trade-off between the two extreme cases of clipping: by increasing $\alpha$, the curve approaches the vanilla clipping method, which results in a larger magnitude after aggregating the clipped gradients at the expense of higher direction deviation. Conversely, decreasing $\alpha$ leads to a better approximation of equal scaling, and thus less direction deviation, while reducing the magnitude of the sum of the clipped gradients (i.e., larger magnitude deviation).

**Illustrative Example** Before developing our differentially private training algorithm via AdaSig, we provide a numerical example to illustrate AdaSig's capability to balance direction and magnitude deviations. Let's consider sample gradients as

$$\mathbf{g}_1 = \begin{bmatrix} 0.3 \\ 0.3 \end{bmatrix}, \ \mathbf{g}_2 = \begin{bmatrix} -0.08 \\ 0.05 \end{bmatrix}, \tag{7}$$

that are shown by solid blue arrows in Figure 2. The sum vector, i.e., true batch gradient, is shown with a dashed blue arrow. The clipped gradients using the vanilla method with $C = 0.1$, and their aggregation are shown in orange by solid and dashed arrows, respectively. Comparing the sum of the clipped gradients with the true batch gradient, we observe a notable direction deviation.

We next consider AdaSig clipping with the same threshold $C = 0.1$ and two different choices of the saturation slope, namely, $\alpha = 15$ and $\alpha = 8$. The result for $\alpha = 15$ is shown by green arrows with the dashed one indicating the sum of the clipped gradients. Evidently, the direction deviation is slightly reduced in this case compared with the vanilla method. As we reduce $\alpha$ to $\alpha = 8$, shown by red arrows, the direction deviation is further reduced. It is however observed that better alignment in this case is achieved at the cost of a smaller magnitude of the aggregated vector. This explains the natural trade-off between direction deviation and magnitude deviation. Appendix H.1 provides additional insights into the effect of $\alpha$ on this trade-off.

## 5    DP-SGD with AdaSig Clipping

We develop a DP-SGD algorithm with AdaSig clipping, which we refer to as DP-SGD-AdaSig. Nonetheless, AdaSig can be readily incorporated into other differentially private optimizers. DP-

SGD-AdaSig consists of two steps: (i) Differentially private update of the model parameters, and (ii) Differentially private update of the saturation slope $\alpha$.

## 5.1 Differentially Private Update of Model Parameters

Rewriting AdaSig clipping in terms of $\alpha_t$ in (6), we have

$$\tilde{\mathbf{g}}_{i,t} = \psi_{\alpha_t}\big(\|\mathbf{g}_{i,t}\|\big)\frac{\mathbf{g}_{i,t}}{\|\mathbf{g}_{i,t}\|}, \tag{8}$$

where we add the subscript $t$ to the slope, i.e., $\alpha_t$, to indicate that it is an adjustable parameter and changes over iterations. We further define $s(\boldsymbol{\theta}_t, \alpha_t; \mathcal{B}_t)$ as the aggregation of clipped gradients in iteration $t$, and use $\hat{s}(\boldsymbol{\theta}_t, \alpha_t; \mathcal{B}_t)$ to represent the private estimation of batch gradient, i.e.,

$$s(\boldsymbol{\theta}_t, \alpha_t; \mathcal{B}_t) \triangleq \sum_{i \in \mathcal{B}_t} \psi_{\alpha_t}\big(\|\mathbf{g}_{i,t}\|\big)\frac{\mathbf{g}_{i,t}}{\|\mathbf{g}_{i,t}\|}, \tag{9a}$$

$$\hat{s}(\boldsymbol{\theta}_t, \alpha_t; \mathcal{B}_t) \triangleq s(\boldsymbol{\theta}_t, \alpha_t; \mathcal{B}_t) + \mathbf{n}_t^s, \tag{9b}$$

where $\mathbf{n}_t^s \sim \mathcal{N}(0, C^2\sigma_s^2\mathbf{I}_d)$ is privacy noise with per-coordinate variance $C^2\sigma_s^2$. It is worth mentioning that $s(\boldsymbol{\theta}_t, \alpha_t; \mathcal{B}_t)$ includes $\boldsymbol{\theta}_t$ as an argument, since it depends on $\boldsymbol{\theta}_t$ through sample gradient $\mathbf{g}_{i,t}$ computed at $\boldsymbol{\theta}_t$.

Using the above notation, the differentially private SGD update of model parameters $\boldsymbol{\theta}_t$ is

$$\boldsymbol{\theta}_{t+1} = \boldsymbol{\theta}_t - \frac{\lambda}{B}\hat{s}(\boldsymbol{\theta}_t, \alpha_t; \mathcal{B}_t). \tag{10}$$

Note that in (10), the parameter $C$ can be absorbed into the learning rate $\lambda$, since $\hat{s}(\boldsymbol{\theta}_t, \alpha_t; \mathcal{B}_t)$ linearly scales with $C$. In fact, the noise term $\mathbf{n}_t^s$ and all clipped gradients in $s(\boldsymbol{\theta}_t, \alpha_t; \mathcal{B}_t)$ are scaled by $C$ (see (5), and note that the variance of $\mathbf{n}_t^s$ is $C^2\sigma_s^2$). We hence simplify the parameter space in the remainder of this work by setting $C = 1$.

## 5.2 Differentially Private Update of Saturation Slope

At iteration $t$, we aim to update $\alpha_t$ to minimize the empirical loss. To this end, we update $\alpha$ in the descent direction of the loss gradient. Developing a differentially private update rule for $\alpha_t$ therefore involves two main steps: (i) computing the derivative of the loss w.r.t. $\alpha$, and (ii) obtaining a differentially private estimate of this derivative. Based on this estimate, a differentially private update rule can then be constructed using one-step gradient descent. We detail these steps in the subsequent sections.

**Loss Derivative with Respect to** $\alpha$ Let $\frac{\partial h^i(\boldsymbol{\theta}_t)}{\partial \alpha_t}$ denote the derivative of the loss on sample $i$ w.r.t. $\alpha$ at iteration $t$. We approximate it as

$$\frac{\partial h^i(\boldsymbol{\theta}_t)}{\partial \alpha_t} \approx -\frac{\lambda}{B}\mathbf{g}_{i,t}^\mathsf{T} r(\boldsymbol{\theta}_{t-1}, \alpha_{t-1}; \mathcal{B}_{t-1}), \tag{11}$$

where

$$r(\boldsymbol{\theta}_t, \alpha_t; \mathcal{B}_t) \triangleq \sum_{i \in \mathcal{B}_t} \frac{2\mathrm{e}^{-\alpha_t\|\mathbf{g}_{i,t}\|}\mathbf{g}_{i,t}}{(1 + \mathrm{e}^{-\alpha_t\|\mathbf{g}_{i,t}\|})^2}. \tag{12}$$

The derivation of the approximation in (11) is provided in Appendix C.

Let $\frac{\partial h(\boldsymbol{\theta}_t)}{\partial \alpha_t}$ denote the batch-averaged loss derivative w.r.t. $\alpha$ at iteration $t$. We approximate it as

$$\begin{aligned}\frac{\partial h(\boldsymbol{\theta}_t)}{\partial \alpha_t} &= \frac{1}{B}\sum_{i \in \mathcal{B}_t}\frac{\partial h^i(\boldsymbol{\theta}_t)}{\partial \alpha_t} \\ &\overset{(a)}{\approx} -\frac{\lambda}{B^2}\Big(\sum_{i \in \mathcal{B}_t}\mathbf{g}_{i,t}\Big)^\mathsf{T} r(\boldsymbol{\theta}_{t-1}, \alpha_{t-1}; \mathcal{B}_{t-1}),\end{aligned} \tag{13}$$

where (a) applies the approximation in (11).

**Differentially Private Estimate of Loss Derivative** We next compute a differentially private estimate of the loss derivative approximation in (13). The right-hand side (RHS) of (13) comprises two factors. For the first factor, i.e., $\sum_{i \in \mathcal{B}_t} \mathbf{g}_{i,t}$, a private estimate is given by $\hat{s}(\boldsymbol{\theta}_t, \alpha_t; \mathcal{B}_t)$, which is computed for update of $\boldsymbol{\theta}_t$ in (10). We can hence obtain a differentially private estimate for $\frac{\partial h(\boldsymbol{\theta}_t)}{\partial \alpha_t}$ by finding a private estimate of the second factor, i.e., $r(\boldsymbol{\theta}_{t-1}, \alpha_{t-1}; \mathcal{B}_{t-1})$. The standard approach for differentially private estimation is to (i) restrict the $\ell_2$-sensitivity of $r(\boldsymbol{\theta}_t, \alpha_t; \mathcal{B}_t)$, and (ii) add a privacy noise term, whose standard deviation is proportional to the bounded $\ell_2$-sensitivity. However, the former step is not required in this case, as the $\ell_2$-sensitivity of $r(\boldsymbol{\theta}_t, \alpha_t; \mathcal{B}_t)$ is bounded. This is shown in the following lemma.

**Lemma 5.1.** *The $\ell_2$-sensitivity of $r(\boldsymbol{\theta}_t, \alpha_t; \mathcal{B}_t)$ is bounded from above by $\Delta_{\alpha_t} = 0.448/\alpha_t$.*

*Proof.* See Appendix D. $\qquad\square$

Using Lemma 5.1, we compute the private estimate of $r(\boldsymbol{\theta}_t, \alpha_t; \mathcal{B}_t)$ as

$$\hat{r}(\boldsymbol{\theta}_t, \alpha_t; \mathcal{B}_t) = r(\boldsymbol{\theta}_t, \alpha_t; \mathcal{B}_t) + \mathbf{n}_t^r, \tag{14}$$

where $\mathbf{n}_t^r \sim \mathcal{N}(0, \Delta_{\alpha_t}^2 \sigma_r^2 \mathbf{I}_d)$ is the Gaussian privacy noise with the noise multiplier $\sigma_r$. Replacing the first and second factors in (13) with their private estimates $\hat{s}(\boldsymbol{\theta}_t, \alpha_t; \mathcal{B}_t)$ and $\hat{r}(\boldsymbol{\theta}_{t-1}, \alpha_{t-1}; \mathcal{B}_{t-1})$, respectively, we obtain the differentially private update for $\alpha_t$ as

$$\alpha_{t+1} = \alpha_t + \frac{\lambda_\alpha \lambda}{B^2} \hat{s}(\boldsymbol{\theta}_t, \alpha_t; \mathcal{B}_t)^\mathsf{T} \hat{r}(\boldsymbol{\theta}_{t-1}, \alpha_{t-1}; \mathcal{B}_{t-1}), \tag{15}$$

for some learning rate $\lambda_\alpha$. Since both $\hat{s}(\boldsymbol{\theta}_t, \alpha_t; \mathcal{B}_t)$ and $\hat{r}(\boldsymbol{\theta}_{t-1}, \alpha_{t-1}; \mathcal{B}_{t-1})$ are noisy, their product can deviate notably from the derivative approximation in (13). To make the update rule robust, we consider only the sign of their product, which is more resilient to noise. Also, to ensure that the value of $\alpha_t$ remains positive, we adopt a standard exponential update, by rewriting (15) in the exponential form as

$$\alpha_{t+1} = \alpha_t \cdot \mathrm{e}^{\lambda_\alpha \mathrm{sign}\left(\hat{s}(\boldsymbol{\theta}_t, \alpha_t; \mathcal{B}_t)^\mathsf{T} \hat{r}(\boldsymbol{\theta}_{t-1}, \alpha_{t-1}; \mathcal{B}_{t-1})\right)}. \tag{16}$$

The DP-SGD-AdaSig algorithm is outlined in Algorithm 1. DP-SGD-AdaSig incurs negligible extra computational and memory costs compared with the vanilla clipping method. A complexity analysis is provided in Appendix B.

---

**Algorithm 1** DP-SGD-AdaSig

1: **Input:** $\mathcal{D}$, $T$, $\alpha_0 > 0$, $\lambda_\alpha$, $\lambda$, $B$, $\sigma_s$, $\sigma_r$, $\boldsymbol{\theta}_0$
2: **Output:** $\{\boldsymbol{\theta}_t\}_{t=1}^T$
3: $\hat{r}(\boldsymbol{\theta}_{-1}, \alpha_{-1}; \mathcal{B}_{-1}) = \mathbf{0}$          // Initialization
4: **for** $t \in \{0, ..., T-1\}$ **do**
5:      Form $\mathcal{B}_t$ via Poisson sampling with rate $B/N$.      // Poisson sampling
6:      **for** $(x_i, y_i) \in \mathcal{B}_t$ **in parallel do**
7:          $\tilde{\mathbf{g}}_{i,t} = \psi_{\alpha_t}(\|\mathbf{g}_{i,t}\|) \frac{\mathbf{g}_{i,t}}{\|\mathbf{g}_{i,t}\|}$
8:          $\mathbf{p}_{i,t} = \frac{2\mathrm{e}^{-\alpha_t \|\mathbf{g}_{i,t}\|} \mathbf{g}_{i,t}}{(1+\mathrm{e}^{-\alpha_t \|\mathbf{g}_{i,t}\|})^2}$
9:      **end for**
10:     $\Delta_{\alpha_t} = \frac{0.448}{\alpha_t}$          // Sensitivity bound
11:     $\hat{s}(\boldsymbol{\theta}_t, \alpha_t; \mathcal{B}_t) = \sum_{i \in \mathcal{B}_t} \tilde{\mathbf{g}}_{i,t} + \mathcal{N}(0, \sigma_s^2 \mathbf{I}_d)$
12:     $\hat{r}(\boldsymbol{\theta}_t, \alpha_t; \mathcal{B}_t) = \sum_{i \in \mathcal{B}_t} \mathbf{p}_{i,t} + \mathcal{N}(0, \Delta_{\alpha_t}^2 \sigma_r^2 \mathbf{I}_d)$
13:     $\boldsymbol{\theta}_{t+1} = \boldsymbol{\theta}_t - \frac{\lambda}{B} \hat{s}(\boldsymbol{\theta}_t, \alpha_t; \mathcal{B}_t)$          // Model update
14:     $\alpha_{t+1} = \alpha_t \cdot \mathrm{e}^{\lambda_\alpha \mathrm{sign}\left(\hat{s}(\boldsymbol{\theta}_t, \alpha_t; \mathcal{B}_t)^\mathsf{T} \hat{r}(\boldsymbol{\theta}_{t-1}, \alpha_{t-1}; \mathcal{B}_{t-1})\right)}$      // $\alpha$ update
15: **end for**

---

## 6 Privacy Analysis of DP-SGD-AdaSig

The privacy guarantee of the proposed algorithm is provided directly by extending the results for DP-SGD. To this end, we first recall that each iteration of the DP-SGD algorithm is a Gaussian

mechanism. The entire DP-SGD algorithm is then viewed as a composition of these Gaussian mechanisms, where each iteration's output serves as the input for the next iteration. The privacy accountant method is then used to obtain the noise multiplier $\sigma$ in (4), which ensures the algorithm satisfies $(\epsilon, \delta)$-DP after $T$ iterations [1]. We denote this guarantee as $\sigma = \texttt{Accountant}(q, T, \epsilon, \delta)$, where $q = B/N$ is the batch sampling probability.

Unlike the DP-SGD algorithm, DP-SGD-AdaSig in each iteration consists of two parallel Gaussian mechanisms: the first one returns the Gaussian approximation of the batch gradient, $\hat{s}(\boldsymbol{\theta}_t, \alpha_t; \mathcal{B}_t)$, that is used for updating both the model in (10) and the saturation slope $\alpha_t$ in (16). The second mechanism returns the Gaussian approximation of $r(\boldsymbol{\theta}_{t-1}, \alpha_{t-1}; \mathcal{B}_{t-1})$ denoted by $\hat{r}(\boldsymbol{\theta}_{t-1}, \alpha_{t-1}; \mathcal{B}_{t-1})$, which is used to update $\alpha_t$ in (16). From a privacy perspective, these two parallel Gaussian mechanisms together behave as a single Gaussian mechanism [3, Theorem 1]. This is formalized in Proposition 6.1.

**Proposition 6.1.** *In each iteration of DP-SGD-AdaSig, the two parallel Gaussian mechanisms $\hat{s}(\cdot)$ and $\hat{r}(\cdot)$, with noise multipliers $\sigma_s$ and $\sigma_r$, are equivalent to a single Gaussian mechanism with noise multiplier $\sigma = (\sigma_r^{-2} + \sigma_s^{-2})^{-1/2}$. Hence, each iteration of DP-SGD-AdaSig incurs the same privacy cost as an iteration of DP-SGD with noise multiplier $\sigma$.*

Compared with Theorem 1 in [3], Proposition 6.1 uses $\sigma_r$ instead of $2\sigma_r$ for privatizing the second mechanism. This is because the privacy noise of the second mechanism in [3] privatizes a positive scalar, but in our algorithm, the noise privatizes $r(\boldsymbol{\theta}_t, \alpha_t; \mathcal{B}_t)$ which may have non-positive entries.

## 7 Convergence Analysis of DP-SGD-AdaSig

Since the clipping structure of AdaSig is substantially different from established methods in the literature, existing convergence analyses are not directly applicable. Furthermore, the adaptive adjustment of $\alpha$ adds complexity to the analysis of training convergence. In this section, we provide theoretical guarantees on the convergence of DP-SGD-AdaSig. We consider the following assumptions on the population loss $L(\boldsymbol{\theta})$ and its gradient, denoted by $\mathbf{g}_t \triangleq \nabla L(\boldsymbol{\theta}_t)$.

**Assumption 7.1.** (Lower bounded loss) There exists a scalar $L^\star$ such that $L(\boldsymbol{\theta}) \geq L^\star, \forall \boldsymbol{\theta} \in \mathbb{R}^d$.

**Assumption 7.2.** (Smoothness) $L(\boldsymbol{\theta})$ is $\beta$-smooth, i.e., $\exists \beta > 0$ such that

$$L(\boldsymbol{\theta}_1) \leq L(\boldsymbol{\theta}_2) + \left\langle \nabla L(\boldsymbol{\theta}_2), \boldsymbol{\theta}_1 - \boldsymbol{\theta}_2 \right\rangle + \frac{\beta}{2} \|\boldsymbol{\theta}_1 - \boldsymbol{\theta}_2\|^2, \forall \boldsymbol{\theta}_1, \boldsymbol{\theta}_2 \in \mathbb{R}^d. \tag{17}$$

**Assumption 7.3.** (Bounded gradient). The $\ell_2$-norm of the gradient of $L(\boldsymbol{\theta})$ is bounded, i.e., there exists $G > 0$, such that $\|\mathbf{g}_t\| \leq G$ for any $\boldsymbol{\theta}_t \in \mathbb{R}^d$.

**Assumption 7.4.** (Unbiased per-sample gradient). The per-sample gradients, i.e., $\mathbf{g}_{i,t}$ for $i \in \mathcal{B}_t$, are i.i.d. and are unbiased estimators of $\mathbf{g}_t$. This means $\mathbf{g}_{i,t} \sim \mathbf{v}_t, \forall i$, where $\mathbf{v}_t = \mathbf{g}_t + \boldsymbol{\Delta}_t, \forall t$, with $\mathbb{E}[\boldsymbol{\Delta}_t] = \mathbf{0}$. Additionally, $\mathbf{g}_{i,t}$ is distributed centrally symmetric around $\mathbf{g}_t$, i.e., $\boldsymbol{\Delta}_t \overset{\mathcal{D}}{=} -\boldsymbol{\Delta}_t$.

Assumptions 7.1 and 7.2 are standard assumptions used for analyzing the convergence of first-order optimization algorithms [4, 7, 35, 41]. Assumption 7.3 is widely used for analysis of clipping in DP algorithms [32, 35, 41]. Assumption 7.4 (unbiasedness and symmetric distribution) is commonly used in both literature not concerned with DP [6, 30, 33, 38] and DP literature [4]. Specifically, the symmetric per-sample gradient distribution assumption is also empirically verified in [7, Figure 3].

We next present the main convergence result in Theorem 7.5, which shows that DP-SGD-AdaSig yields a bounded weighted average of the expected squared gradient norms. The bound is expressed explicitly in terms of the privacy budget and other parameters.

**Theorem 7.5.** *Let Assumptions 7.1–7.4 hold. For a constant $r \geq G$, let $\alpha_0 \propto 1/r$. Moreover, let $\lambda_\alpha \propto 1/T$ and $\lambda = \sqrt{\frac{2N^2\epsilon^2\left(L(\boldsymbol{\theta}_0) - L^\star\right)}{\beta T\left(2N^2\epsilon^2 + d\nu_s^2 T \log\left(\frac{1}{\delta}\right)\right)}}$. Then, under the privacy budget $(\epsilon, \delta)$, there exist a constant $\nu_s > 0$ such that DP-SGD-AdaSig satisfies the following inequality:*

$$\frac{1}{T} \sum_{t=0}^{T-1} \Pr\left\{ \|\boldsymbol{\Delta}_t\| \leq r \right\} \mathbb{E}\|\mathbf{g}_t\|^2 \leq 2r\tilde{G}\sqrt{\beta\mathcal{J}}, \tag{18}$$

*for some constant $\tilde{G} > 0$, where*

$$\mathcal{J} \triangleq \frac{L(\boldsymbol{\theta}_0) - L^\star}{T}\left(1 + \frac{d\nu_s^2 T \log(\frac{1}{\delta})}{2N^2\epsilon^2}\right). \tag{19}$$

*Proof.* See Appendix E.2. □

We note regarding the bound in Theorem 7.5: (i) The weight of expected gradient norm squared in the expression on the left-hand side (LHS) of (18) corresponds to the probability that the magnitude of the deviation of sample gradients from $\mathbf{g}_t$ does not exceed $r$. (ii) Tighter privacy guarantees, i.e., smaller values of $(\epsilon, \delta)$, lead to higher values on the RHS. This reflects the trade-off between privacy and learning performance. In Corollary 7.6 below, we observe that DP-SGD-AdaSig achieves the asymptotic convergence rate of $\mathcal{O}(\frac{\sqrt{d}}{N\epsilon})$, which matches the known privacy–utility trade-off bound in the literature for vanilla clipping that does not allow adaptation [7, 12].

**Corollary 7.6.** *As $T$ approaches infinity, the RHS of* (18) *converges to* $\mathcal{O}(\frac{\sqrt{d\log(1/\delta)}}{N\epsilon})$.

When the sample gradients have a bounded variance, the weights in the LHS of the expression in Theorem 7.5 can be further simplified. This yields an upper bound for the average of the expected gradient norm squared, which is given in the following corollary.

**Corollary 7.7.** *Let $\mathbb{E}\|\boldsymbol{\Delta}_t\|^2 \leq \zeta^2$ for any iteration $t$. Then, the bound in Theorem 7.5 reduces to*

$$\frac{1}{T}\sum_{t=0}^{T-1}\mathbb{E}\|\mathbf{g}_t\|^2 \leq 2R\tilde{G}\sqrt{\beta\mathcal{J}}, \tag{20}$$

*where $R \triangleq \frac{r_\star^2}{r_\star - \zeta}$, and $r_\star = \max\{2\zeta, G\}$.*

*Proof.* See Appendix E.3. □

## 8 Experiments

We evaluate AdaSig on image and sentence classification tasks.[2] Under equal privacy budget $(\epsilon, \delta)$, we compare the performance of AdaSig against four baselines:[3] (i) Vanilla method [1], (ii) method proposed in [3], (iii) Auto-S [4], and (iv) PSAC [36]. We further present an ablation study for AdaSig in Appendix H, where we compare the performance of AdaSig with sigmoid clipping using a fixed $\alpha$.

### 8.1 Image Classification Task

We conduct experiments on five image classification datasets including MNIST [20], FashionMNIST [37], CIFAR-10 [18], ImageNette [15] (a 10-class subset of ImageNet [9]), and CelebA [23].

**Setting** For MNIST, FashionMNIST and CIFAR-10, we train the CNN architectures used in [27, 34, 4], i.e., a 4-layer CNN for MNIST and FashionMNIST, and an 8-layer CNN for CIFAR-10. The simulation setup and hyperparameters are adopted from [34]. For ImageNette, we use ResNet-9 with group normalization (instead of batch normalization) and *without* scale normalization, following the setup considered in [17, 4], with the minor exception that the learning rate does not decay. For CelebA, we use the same ResNet-9 architecture as for ImageNette, with the same simulation setup as in [4]. Since each image in CelebA dataset has 40 labels, we run two sets of experiments on this dataset: (i) We consider a binary classification problem that aims to predict the `'Smiling'` label only. (ii) We consider the problem of multi-label classification with all the available 40 labels in the dataset. The detailed settings, including hyperparameters for each method, are provided in Appendix F.2.

**Performance Comparison** Table 1 (first six rows) shows the average test accuracy across five different runs of each dataset. AdaSig outperforms all baselines by achieving higher average accuracies

---

[2]The GitHub repository for our implementation is available at `https://github.com/faezemoradik/AdaptiveSigmoidClipping.git`.

[3]Methods such as [41] are not directly comparable, as they do not propose a new clipping strategy but instead incorporate error feedback into existing ones, which can be applied to any clipping method, including AdaSig.

Table 1: Average test accuracy (in percentages) with 95% confidence intervals on five runs. Bold numbers indicate $p$-value $< 0.05$ in the t-test performed to identify statistical significance.

| DATASET | MODEL | $(\epsilon, \delta)$ | VANILLA | METHOD IN [3] | AUTO-S | PSAC | ADASIG |
|---------|-------|------|---------|---------------|--------|------|--------|
| MNIST | 4-LAYER CNN | $(3, 10^{-5})$ | $98.11 \pm 0.04$ | $98.13 \pm 0.03$ | $98.04 \pm 0.04$ | $98.12 \pm 0.04$ | $98.14 \pm 0.05$ |
| FASHIONMNIST | 4-LAYER CNN | $(3, 10^{-5})$ | $86.27 \pm 0.24$ | $86.28 \pm 0.19$ | $86.20 \pm 0.26$ | $86.35 \pm 0.19$ | $\mathbf{86.68 \pm 0.04}$ |
| CIFAR-10 | 8-LAYER CNN | $(3, 10^{-5})$ | $61.49 \pm 0.13$ | $61.53 \pm 0.11$ | $61.21 \pm 0.25$ | $61.93 \pm 0.14$ | $\mathbf{62.54 \pm 0.20}$ |
| IMAGENETTE | RESNET-9 | $(8, 10^{-4})$ | $62.02 \pm 0.91$ | $62.07 \pm 0.78$ | $61.44 \pm 0.69$ | $62.30 \pm 0.42$ | $\mathbf{64.65 \pm 0.34}$ |
| CELEBA (MULTI-LABEL) | RESNET-9 | $(8, 5 \times 10^{-6})$ | $88.51 \pm 0.03$ | $88.53 \pm 0.02$ | $88.48 \pm 0.03$ | $88.46 \pm 0.02$ | $\mathbf{88.78 \pm 0.03}$ |
| CELEBA (SMILING) | RESNET-9 | $(8, 5 \times 10^{-6})$ | $90.87 \pm 0.28$ | $90.88 \pm 0.26$ | $90.73 \pm 0.19$ | $90.90 \pm 0.26$ | $90.95 \pm 0.14$ |
| SST-2 | ROBERTA-BASE | $(3, \frac{1}{2N})$ | $87.43 \pm 0.49$ | $87.46 \pm 0.47$ | $87.18 \pm 0.56$ | $87.48 \pm 0.32$ | $\mathbf{88.44 \pm 0.17}$ |
| QNLI | ROBERTA-BASE | $(3, \frac{1}{2N})$ | $84.32 \pm 0.16$ | $84.36 \pm 0.17$ | $84.51 \pm 0.13$ | $84.58 \pm 0.14$ | $\mathbf{84.92 \pm 0.12}$ |

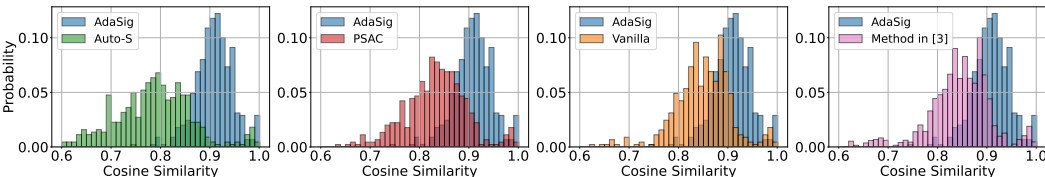

Figure 3: Comparison of cosine similarity during training ResNet-9 on ImageNette.

across all datasets. Conducting a statistical t-test further confirms the statistical significance of this improvement ($p$-value $< 0.05$) on 4 out of 6 datasets [14].

**Cosine Similarity Analysis** To assess AdaSig's ability to balance the direction deviation and magnitude deviation, we measure the cosine similarity between the true batch gradient and the aggregation of clipped gradients. The results are shown in Figure 3, where we compare the histogram of the cosine similarity while training ResNet-9 on ImageNette. Notably, AdaSig's histogram is concentrated at larger cosine similarities as compared with the baselines, which indicates the effectiveness of AdaSig clipping in reducing the direction deviation. Additional results are presented in Appendix G to elaborate on the trade-off between the direction and magnitude deviations.

## 8.2 Sentence Classification Task

We conduct sentence classification experiments on SST-2 [31], for sentiment classification, and QNLI [28], for question-answering inference both taken from the GLUE benchmark.

**Setting** We use the pre-trained RoBERTa-base model and conduct full parameter fine-tuning with DP consideration following the same setting as in [21, 4]. The detailed settings, including hyperparameters for each method, are provided in Appendix F.3.

**Performance Comparison** Table 1 (last two rows) shows the average test accuracy over five runs for each dataset. We observe that AdaSig achieves statistically significantly ($p$-value $< 0.05$) higher average test accuracies as compared with all baselines on both datasets.

## 9 Conclusions

We proposed AdaSig clipping, a novel clipping operation for differentially private training that balances the trade-off between direction and magnitude deviations incurred by clipping. It achieves this balance by adaptively adjusting its saturation slope throughout training based on information obtained from sample gradients. Our convergence analysis demonstrates that DP-SGD with AdaSig clipping retains the best-known convergence rate in the non-convex loss setting. Experiments on image and sentence classification tasks demonstrate that AdaSig clipping consistently improves training performance compared to existing clipping methods.

## Acknowledgments

This work was supported in part by the Natural Sciences and Engineering Research Council of Canada.

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

# APPENDIX

The appendix is structured as follows: Appendix A presents details on the choice of the sigmoid function for clipping. The complexity analysis for DP-SGD-AdaSig is given in Appendix B. An approximation of the sample loss derivative is provided in Appendix C. The proof of Lemma 5.1 is presented in Appendix D. The proof of Theorem 7.5 and the preliminary lemmas and theorems related to the convergence analysis are included in Appendix E. The detailed experimental setup is presented in Appendix F. A numerical analysis of the direction and magnitude deviations is provided in Appendix G. The ablation study of AdaSig is presented in Appendix H. Finally, Appendix I and Appendix J report the variation of $\alpha_t$ across training iterations and the numerical convergence results, respectively.

## A    Details on the Choice of Sigmoid Function for Clipping

The sigmoid function is simple yet well-suited to our goal of adaptively balancing the trade-off between direction and magnitude deviations. In particular, it enables control over the range of the linear span in Figure 1 through the parameter $\alpha$. Although PSAC [36] and Auto-S [4] could be extended to variants that adjust their linear region span by varying the constant parameter $r$ throughout training, the sigmoid function provides greater flexibility and control.

Specifically, in the PSAC clipping function, varying $r$ alters the span of its linear region; however, the change in this span is confined to a limited range. This behavior can be illustrated by analyzing the PSAC clipping function, $\tilde{\mathbf{g}}_{i,t} = \frac{C\mathbf{g}_{i,t}}{\|\mathbf{g}_{i,t}\| + \frac{r}{\|\mathbf{g}_{i,t}\|+r}}$, in extreme cases. Due to the term $\frac{r}{\|\mathbf{g}_{i,t}\|+r}$ in the denominator, varying $r$ from 0 to $\infty$ causes the clipped gradient norm curve $\|\tilde{\mathbf{g}}_{i,t}\|$ (shown in Figure 1) to transition only between two limiting curves: $C$ and $\frac{C\|\mathbf{g}_{i,t}\|}{\|\mathbf{g}_{i,t}\|+1}$. Consequently, the span of the linear region in PSAC is restricted to lie between these two curves, and adaptively updating $r$ cannot effectively balance the trade-off between direction and magnitude deviations. In contrast, in the AdaSig clipping function of (6), the clipped gradient norm curve $\|\tilde{\mathbf{g}}_{i,t}\|$ (illustrated in Figure 1) can vary from $C$ (as $\alpha \to \infty$) down to values arbitrarily close to 0 (as $\alpha \to 0$), thereby providing greater flexibility in adjusting the span of the linear region.

While the Auto-S clipping function, $\tilde{\mathbf{g}}_{i,t} = \frac{C\mathbf{g}_{i,t}}{\|\mathbf{g}_{i,t}\|+r}$, does not have the restricted variation of the linear region span seen in PSAC, it is still less flexible than the sigmoid function. Its linear region span is less responsive to changes in $r$ than the sigmoid function is to $\alpha$, so larger adjustments of $r$ are required to produce a noticeable effect. In contrast, the sigmoid function, with its exponential dependence on $\alpha$, offers more flexible control over the linear region and thus the direction–magnitude mismatch trade-off.

## B    Complexity Analysis

Compared with DP-SGD (with vanilla clipping), one can observe that DP-SGD-AdaSig (Algorithm 1) introduces a negligible increase in computational and memory costs. Specifically, with respect to computation, the algorithm performs lines 8, 12, and 14 in addition to the standard operations in vanilla clipping. Line 8 is carried out efficiently using the per-sample gradient norms computed in line 7, the computational cost of step 12 is identical to that of step 11, and line 14 incurs only a constant additional cost per iteration. Consequently, the overall computational cost introduced by these additional steps is dominated by other operations and the cost of per-sample gradient computation, making the additional overall computational cost minimal.

With respect to memory, DP-SGD-AdaSig requires extra memory to store $\hat{r}(\boldsymbol{\theta}_t, \alpha_t; \mathcal{B}_t)$ (line 12), which is required for the slope update in the next iteration (line 14). This additional memory cost is negligible as compared with the memory required for storing the model parameters and per-sample gradients.

## C  Approximation of the Sample Loss Derivative with Respect to $\alpha$

Since the loss on sample $i$ in iteration $t$, denoted by $h^i(\boldsymbol{\theta}_t)$, does not directly depend on $\alpha_t$, to derive an update rule for $\alpha$, we unroll the expression for $h^i(\boldsymbol{\theta}_t)$ and write it in terms of $\boldsymbol{\theta}_{t-1}$ and $\alpha_{t-1}$. In particular, using (10), we obtain

$$h^i(\boldsymbol{\theta}_t) = h^i\left(\boldsymbol{\theta}_{t-1} - \frac{\lambda}{B}\hat{s}(\boldsymbol{\theta}_{t-1}, \alpha_{t-1}; \mathcal{B}_{t-1})\right). \tag{21}$$

For sufficiently small learning rate for $\alpha$ (i.e., $\lambda_\alpha$ in (16)), $\alpha_{t-1}$ and $\alpha_t$ differ only slightly, and hence we adopt the approximation $\alpha_{t-1} \approx \alpha_t$. Consequently, the loss derivative at $\alpha_t$, $\frac{\partial h^i(\boldsymbol{\theta}_t)}{\partial \alpha_t}$, can be approximated by the loss derivative at $\alpha_{t-1}$, $\frac{\partial h^i(\boldsymbol{\theta}_t)}{\partial \alpha_{t-1}}$, i.e.,

$$\frac{\partial h^i(\boldsymbol{\theta}_t)}{\partial \alpha_t} \approx \frac{\partial h^i(\boldsymbol{\theta}_t)}{\partial \alpha_{t-1}}. \tag{22}$$

We now apply the chain rule to (21) to compute $\frac{\partial h^i(\boldsymbol{\theta}_t)}{\partial \alpha_{t-1}}$ as

$$\frac{\partial h^i(\boldsymbol{\theta}_t)}{\partial \alpha_{t-1}} = \left(\nabla_{\boldsymbol{\theta}_t} h^i(\boldsymbol{\theta}_t)\right)^\mathsf{T} \frac{\partial \boldsymbol{\theta}_t}{\partial \alpha_{t-1}}, \tag{23}$$

where the first factor in the RHS of (23) is the sample loss gradient w.r.t. model parameters, i.e., $\mathbf{g}_{i,t}$. To compute the second factor, we write

$$
\begin{aligned}
\frac{\partial \boldsymbol{\theta}_t}{\partial \alpha_{t-1}} &= \frac{\partial\left(\boldsymbol{\theta}_{t-1} - \frac{\lambda}{B}\hat{s}(\boldsymbol{\theta}_{t-1}, \alpha_{t-1}; \mathcal{B}_{t-1})\right)}{\partial \alpha_{t-1}} \\
&\overset{(a)}{=} \frac{-\lambda}{B} \sum_{i \in \mathcal{B}_{t-1}} \frac{\partial \psi_{\alpha_{t-1}}(\|\mathbf{g}_{i,t-1}\|)}{\partial \alpha_{t-1}} \frac{\mathbf{g}_{i,t-1}}{\|\mathbf{g}_{i,t-1}\|} \\
&\overset{(b)}{=} \frac{-\lambda}{B} \sum_{i \in \mathcal{B}_{t-1}} \frac{2\mathrm{e}^{-\alpha_{t-1}\|\mathbf{g}_{i,t-1}\|}\mathbf{g}_{i,t-1}}{(1 + \mathrm{e}^{-\alpha_{t-1}\|\mathbf{g}_{i,t-1}\|})^2},
\end{aligned} \tag{24}
$$

where (a) follows from the definition of $\hat{s}(\boldsymbol{\theta}_{t-1}, \alpha_{t-1}; \mathcal{B}_{t-1})$ in (9b), and (b) is obtained by computing the derivative of the AdaSig function $\psi_{\alpha_{t-1}}(\cdot)$. Using the definition of $r(\boldsymbol{\theta}_t, \alpha_t; \mathcal{B}_t)$ in (12), we rewrite (24) as

$$\frac{\partial \boldsymbol{\theta}_t}{\partial \alpha_{t-1}} = \frac{-\lambda}{B} r(\boldsymbol{\theta}_{t-1}, \alpha_{t-1}; \mathcal{B}_{t-1}). \tag{25}$$

Substituting (25) in (23), we finally get

$$\frac{\partial h^i(\boldsymbol{\theta}_t)}{\partial \alpha_{t-1}} = -\frac{\lambda}{B}\mathbf{g}_{i,t}^\mathsf{T} r(\boldsymbol{\theta}_{t-1}, \alpha_{t-1}; \mathcal{B}_{t-1}). \tag{26}$$

Using (22) together with (26), we have

$$\frac{\partial h^i(\boldsymbol{\theta}_t)}{\partial \alpha_t} \approx -\frac{\lambda}{B}\mathbf{g}_{i,t}^\mathsf{T} r(\boldsymbol{\theta}_{t-1}, \alpha_{t-1}; \mathcal{B}_{t-1}). \tag{27}$$

## D  Proof of Lemma 5.1: Sensitivity Bound

*Proof.* The $\ell_2$-sensitivity of $r(\boldsymbol{\theta}_t, \alpha_t; \mathcal{B}_t)$ w.r.t. the training samples in iteration $t$ is defined as

$$\Delta r = \max_{\mathcal{B}_t, \mathcal{B}_t'} \left\| r(\boldsymbol{\theta}_t, \alpha_t; \mathcal{B}_t) - r(\boldsymbol{\theta}_t, \alpha_t; \mathcal{B}_t') \right\|_2, \tag{28}$$

where $\mathcal{B}_t$ and $\mathcal{B}_t'$ are two adjacent batches of data that differ by exactly one data point. Without loss of generality, we assume that $\mathcal{B}_t'$ contains all elements of $\mathcal{B}_t$ together with an additional data point $(x', y')$, where $x'$ is the input feature and $y'$ is its corresponding target output. That is, $\mathcal{B}_t' = \mathcal{B}_t \cup \{(x', y')\}$. Moreover, let $\mathbf{g}'$ denote the gradient of sample loss at $(x', y')$, i.e., $\mathbf{g}' = \nabla_\theta \ell(f_{\boldsymbol{\theta}}(x'), y')$.

Considering the definition of $r(\boldsymbol{\theta}_t, \alpha_t; \mathcal{B}_t)$ in (12), we can conclude that the terms under summation in $r(\boldsymbol{\theta}_t, \alpha_t; \mathcal{B}_t)$ and $r(\boldsymbol{\theta}_t, \alpha_t; \mathcal{B}_t')$ are identical, except a single term that arises due to the extra element in $\mathcal{B}_t'$. The $\ell_2$-sensitivity in (28) can hence be upper bounded by the norm of this extra term, i.e.,

$$\Delta r \leq \max_{\mathbf{g}'} \left\| \frac{2\mathrm{e}^{-\alpha_t \|\mathbf{g}'\|} \mathbf{g}'}{(1 + \mathrm{e}^{-\alpha_t \|\mathbf{g}'\|})^2}) \right\|_2 \tag{29a}$$

$$= \max_{\mathbf{g}'} \frac{2\mathrm{e}^{-\alpha_t \|\mathbf{g}'\|} \|\mathbf{g}'\|}{(1 + \mathrm{e}^{-\alpha_t \|\mathbf{g}'\|})^2} \tag{29b}$$

$$\overset{(a)}{=} \frac{1}{\alpha_t} \max_{z \geq 0} \frac{2\mathrm{e}^{-z} z}{(1 + \mathrm{e}^{-z})^2}, \tag{29c}$$

where (a) follows from the variable exchange $z \triangleq \alpha_t \|\mathbf{g}'\|$. We next find the solution to the maximization in (29c) by setting the derivative of the objective to zero, i.e., $\frac{\partial}{\partial z} \frac{2\mathrm{e}^{-z} z}{(1+\mathrm{e}^{-z})^2} = 0$, which results in the following fixed-point equation as

$$z = \ln \frac{z+1}{z-1}. \tag{30}$$

Let's denote the solution for (30) by $z_\star$. Substituting the solution, i.e., $z_\star$, into objective (29c), we get the upper bound for $\Delta r$ as

$$\Delta r \leq \frac{1}{\alpha_t} \frac{2\mathrm{e}^{-z_\star} z_\star}{(1 + \mathrm{e}^{-z_\star})^2} = \frac{z_\star^2 - 1}{2z_\star \alpha_t}. \tag{31}$$

Solving (30) numerically for $z$ and substituting $z_\star$ in (31) results in $\Delta r \leq 0.448/\alpha_t$, which completes the proof. $\qquad \square$

# E   Theoretical Convergence Analysis

We first introduce the preliminary lemmas and theorems required for proving the main theorem.

## E.1   Preliminaries

**Lemma E.1.** *Suppose that Assumptions 7.2 and 7.4 hold. Under Algorithm 1, the expectation of the population loss difference in two consecutive iterations is upper-bounded by*

$$\mathbb{E}\Big[L(\boldsymbol{\theta}_{t+1}) - L(\boldsymbol{\theta}_t)\Big|\boldsymbol{\theta}_t, \alpha_t\Big] \leq -\lambda \mathbf{g}_t^\mathsf{T} \mathbb{E}\Big[\Big(\frac{2}{1 + \mathrm{e}^{-\alpha_t \|\mathbf{v}_t\|}} - 1\Big)\frac{\mathbf{v}_t}{\|\mathbf{v}_t\|}\Big|\boldsymbol{\theta}_t, \alpha_t\Big] + \beta\lambda^2\Big(1 + \frac{d\sigma_s^2}{2B^2}\Big), \tag{32}$$

*where $\mathbf{g}_t = \nabla L(\boldsymbol{\theta}_t)$, $\mathbb{E}[\cdot|\boldsymbol{\theta}_t, \alpha_t]$ denote the expectation over the randomness in iteration $t$ for given $\boldsymbol{\theta}_t$ and $\alpha_t$, and $\mathbf{v}_t$ is the random process from which $\mathbf{g}_{i,t}$ is sampled, i.e., $\mathbf{g}_{i,t} \sim \mathbf{v}_t$ for $i \in \mathcal{B}_t$.*

*Proof.* Based on Assumption 7.2, the population loss is $\beta$-smooth. Thus, we have

$$L(\boldsymbol{\theta}_{t+1}) - L(\boldsymbol{\theta}_t) \leq \mathbf{g}_t^\mathsf{T}(\boldsymbol{\theta}_{t+1} - \boldsymbol{\theta}_t) + \frac{\beta}{2}\|\boldsymbol{\theta}_{t+1} - \boldsymbol{\theta}_t\|^2. \tag{33}$$

Taking expectation from both sides of (33) for given $\boldsymbol{\theta}_t$ and $\alpha_t$, we have

$$\mathbb{E}\Big[L(\boldsymbol{\theta}_{t+1}) - L(\boldsymbol{\theta}_t)\Big|\boldsymbol{\theta}_t, \alpha_t\Big]$$

$$\leq \mathbf{g}_t^\mathsf{T}\mathbb{E}\Big[(\boldsymbol{\theta}_{t+1} - \boldsymbol{\theta}_t)\Big|\boldsymbol{\theta}_t, \alpha_t\Big] + \frac{\beta}{2}\mathbb{E}\Big[\|\boldsymbol{\theta}_{t+1} - \boldsymbol{\theta}_t\|^2\Big|\boldsymbol{\theta}_t, \alpha_t\Big] \tag{34}$$

$$\overset{(a)}{=} -\frac{\lambda}{B}\mathbf{g}_t^\mathsf{T}\mathbb{E}\Big[\sum_{i \in \mathcal{B}_t}\psi_{\alpha_t}(\|\mathbf{g}_{i,t}\|)\frac{\mathbf{g}_{i,t}}{\|\mathbf{g}_{i,t}\|} + \mathbf{n}_t^s\Big|\boldsymbol{\theta}_t, \alpha_t\Big]$$
$$+ \frac{\beta\lambda^2}{2B^2}\mathbb{E}\Big[\Big\|\sum_{i \in \mathcal{B}_t}\psi_{\alpha_t}(\|\mathbf{g}_{i,t}\|)\frac{\mathbf{g}_{i,t}}{\|\mathbf{g}_{i,t}\|} + \mathbf{n}_t^s\Big\|^2\Big|\boldsymbol{\theta}_t, \alpha_t\Big] \tag{35}$$

$$\overset{(b)}{=} -\frac{\lambda}{B}\mathbf{g}_t^\mathsf{T}\mathbb{E}\Big[\sum_{i \in \mathcal{B}_t}\psi_{\alpha_t}(\|\mathbf{g}_{i,t}\|)\frac{\mathbf{g}_{i,t}}{\|\mathbf{g}_{i,t}\|}\Big|\boldsymbol{\theta}_t, \alpha_t\Big]$$
$$+ \frac{\beta\lambda^2}{2B^2}\mathbb{E}\Big[\Big\|\sum_{i \in \mathcal{B}_t}\psi_{\alpha_t}(\|\mathbf{g}_{i,t}\|)\frac{\mathbf{g}_{i,t}}{\|\mathbf{g}_{i,t}\|}\Big\|^2\Big|\boldsymbol{\theta}_t, \alpha_t\Big] + \frac{\beta\lambda^2}{2B^2}\mathbb{E}\Big[\|\mathbf{n}_t^s\|^2\Big|\boldsymbol{\theta}_t, \alpha_t\Big] \tag{36}$$

$$\overset{(c)}{\leq} -\frac{\lambda}{B}\mathbf{g}_t^\mathsf{T}\mathbb{E}\Big[\sum_{i \in \mathcal{B}_t}\psi_{\alpha_t}(\|\mathbf{g}_{i,t}\|)\frac{\mathbf{g}_{i,t}}{\|\mathbf{g}_{i,t}\|}\Big|\boldsymbol{\theta}_t, \alpha_t\Big]$$
$$+ \frac{\beta\lambda^2}{2B^2}\mathbb{E}\Big[\Big(\sum_{i \in \mathcal{B}_t}\Big\|\psi_{\alpha_t}(\|\mathbf{g}_{i,t}\|)\frac{\mathbf{g}_{i,t}}{\|\mathbf{g}_{i,t}\|}\Big\|\Big)^2\Big|\boldsymbol{\theta}_t, \alpha_t\Big] + \frac{\beta\lambda^2}{2B^2}d\sigma_s^2 \tag{37}$$

$$\overset{(d)}{\leq} -\frac{\lambda}{B}\mathbf{g}_t^\mathsf{T}\mathbb{E}\Big[\sum_{i \in \mathcal{B}_t}\psi_{\alpha_t}(\|\mathbf{g}_{i,t}\|)\frac{\mathbf{g}_{i,t}}{\|\mathbf{g}_{i,t}\|}\Big|\boldsymbol{\theta}_t, \alpha_t\Big] + \beta\lambda^2\Big(\frac{\mathbb{E}|\mathcal{B}_t|^2}{2B^2} + \frac{d\sigma_s^2}{2B^2}\Big) \tag{38}$$

$$\overset{(e)}{\leq} -\frac{\lambda}{B}\mathbf{g}_t^\mathsf{T}\mathbb{E}\Big[\sum_{i \in \mathcal{B}_t}\psi_{\alpha_t}(\|\mathbf{g}_{i,t}\|)\frac{\mathbf{g}_{i,t}}{\|\mathbf{g}_{i,t}\|}\Big|\boldsymbol{\theta}_t, \alpha_t\Big] + \beta\lambda^2\Big(1 + \frac{d\sigma_s^2}{2B^2}\Big), \tag{39}$$

where (a) follows from (10), (b) comes from the fact that $\mathbf{g}_{i,t}$ for $i \in \mathcal{B}_t$ are independent of the process $\mathbf{n}_t^s$, and the fact that $\mathbf{n}_t^s$ is zero-mean, (c) is obtained by applying the Triangle inequality to the second term in (36) and substituting the variance of $\mathbf{n}_t^s$, (d) results from the upper bound

$$\sum_{i \in \mathcal{B}_t}\Big\|\psi_{\alpha_t}(\|\mathbf{g}_{i,t}\|)\frac{\mathbf{g}_{i,t}}{\|\mathbf{g}_{i,t}\|}\Big\| \leq |\mathcal{B}_t|, \tag{40}$$

since $\psi_{\alpha_t}(x) \leq 1$ for any $x \in \mathbb{R}$, and (e) follows from the fact that, under Poisson sampling with rate $B/N$, we have

$$\frac{\mathbb{E}\big[|\mathcal{B}_t|^2\big]}{B^2} = \frac{B + 1}{B} - \frac{1}{N} \leq 1 + \frac{1}{B} \leq 2, \tag{41}$$

where the last inequality holds since the expected batch size $B \geq 1$.

Based on Assumption 7.4, the sample gradients are identically distributed, i.e., $\mathbf{g}_{i,t} \sim \mathbf{v}_t, \forall i$. We can hence write

$$\frac{1}{B}\mathbb{E}\Big[\sum_{i \in \mathcal{B}_t}\psi_{\alpha_t}(\|\mathbf{g}_{i,t}\|)\frac{\mathbf{g}_{i,t}}{\|\mathbf{g}_{i,t}\|}\Big|\boldsymbol{\theta}_t, \alpha_t\Big] = \mathbb{E}\Big[\psi_{\alpha_t}(\|\mathbf{v}_t\|)\frac{\mathbf{v}_t}{\|\mathbf{v}_t\|}\Big|\boldsymbol{\theta}_t, \alpha_t\Big]. \tag{42}$$

Substituting into (39) and replacing $\psi_\alpha(\|\mathbf{v}_t\|)$ with its definition, (32) is concluded. $\qquad\square$

**Lemma E.2.** *Under Assumption 7.4, i.e., $\mathbf{v}_t = \mathbf{g}_t + \boldsymbol{\Delta}_t$, the following equality holds:*

$$\mathbf{g}_t^\mathsf{T}\mathbb{E}\Big[\Big(\frac{2}{1 + \mathrm{e}^{-\alpha_t\|\mathbf{v}_t\|}} - 1\Big)\frac{\mathbf{v}_t}{\|\mathbf{v}_t\|}\Big|\boldsymbol{\theta}_t, \alpha_t\Big] = \frac{\|\mathbf{g}_t\|}{2}\mathbb{E}_{s,c}\Big[f(s, c, \gamma_t)\Big|0 \leq c \leq 1\Big], \tag{43}$$

*where $\gamma_t \triangleq \alpha_t\|\mathbf{g}_t\|$, $s \triangleq \frac{\|\boldsymbol{\Delta}_t\|}{\|\mathbf{g}_t\|}$, $c \triangleq \frac{\mathbf{g}_t^\mathsf{T}\boldsymbol{\Delta}_t}{\|\boldsymbol{\Delta}_t\|\|\mathbf{g}_t\|}$, and*

$$f(s, c, \gamma_t) \triangleq \frac{1 + sc}{\sqrt{1 + s^2 + 2sc}}\Big(\frac{2}{1 + \mathrm{e}^{-\gamma_t\sqrt{1+s^2+2sc}}} - 1\Big)$$
$$+ \frac{1 - sc}{\sqrt{1 + s^2 - 2sc}}\Big(\frac{2}{1 + \mathrm{e}^{-\gamma_t\sqrt{1+s^2-2sc}}} - 1\Big). \tag{44}$$

*Proof.* We adopt an approach similar to that proposed in [4]. Let $\mathcal{H}^+$ and $\mathcal{H}^-$ denote the following two halfspaces:

$$\mathcal{H}^+ \triangleq \left\{ \mathbf{o} \in \mathbb{R}^d | \mathbf{g}_t^\mathsf{T} \mathbf{o} \geq 0 \right\}, \tag{45}$$

$$\mathcal{H}^- \triangleq \left\{ \mathbf{o} \in \mathbb{R}^d | \mathbf{g}_t^\mathsf{T} \mathbf{o} \leq 0 \right\}. \tag{46}$$

Using $\mathbf{v}_t = \mathbf{g}_t + \boldsymbol{\Delta}_t$, we can write

$$\mathbf{g}_t^\mathsf{T} \mathbb{E}\Big[ \big( \frac{2}{1 + \mathrm{e}^{-\alpha_t \|\mathbf{v}_t\|}} - 1 \big) \frac{\mathbf{v}_t}{\|\mathbf{v}_t\|} \Big| \boldsymbol{\theta}_t, \alpha_t \Big] \tag{47}$$

$$= \mathbb{E}_{\boldsymbol{\Delta}_t}\Big[ \big( \frac{2}{1 + \mathrm{e}^{-\alpha_t \|\mathbf{g}_t + \boldsymbol{\Delta}_t\|}} - 1 \big) \frac{\mathbf{g}_t^\mathsf{T}(\mathbf{g}_t + \boldsymbol{\Delta}_t)}{\|\mathbf{g}_t + \boldsymbol{\Delta}_t\|} \Big] \tag{48}$$

$$\overset{(a)}{=} \frac{1}{2} \mathbb{E}_{\boldsymbol{\Delta}_t}\Big[ \big( \frac{2}{1 + \mathrm{e}^{-\alpha_t \sqrt{\|\mathbf{g}_t\|^2 + \|\boldsymbol{\Delta}_t\|^2 + 2\mathbf{g}_t^\mathsf{T}\boldsymbol{\Delta}_t}}} - 1 \big) \frac{\mathbf{g}_t^\mathsf{T}\mathbf{g}_t + \mathbf{g}_t^\mathsf{T}\boldsymbol{\Delta}_t}{\sqrt{\|\mathbf{g}_t\|^2 + \|\boldsymbol{\Delta}_t\|^2 + 2\mathbf{g}_t^\mathsf{T}\boldsymbol{\Delta}_t}} \Big| \boldsymbol{\Delta}_t \in \mathcal{H}^+ \Big]$$

$$+ \frac{1}{2} \mathbb{E}_{\boldsymbol{\Delta}_t}\Big[ \big( \frac{2}{1 + \mathrm{e}^{-\alpha_t \sqrt{\|\mathbf{g}_t\|^2 + \|\boldsymbol{\Delta}_t\|^2 + 2\mathbf{g}_t^\mathsf{T}\boldsymbol{\Delta}_t}}} - 1 \big) \frac{\mathbf{g}_t^\mathsf{T}\mathbf{g}_t + \mathbf{g}_t^\mathsf{T}\boldsymbol{\Delta}_t}{\sqrt{\|\mathbf{g}_t\|^2 + \|\boldsymbol{\Delta}_t\|^2 + 2\mathbf{g}_t^\mathsf{T}\boldsymbol{\Delta}_t}} \Big| \boldsymbol{\Delta}_t \in \mathcal{H}^- \Big] \tag{49}$$

$$\overset{(b)}{=} \frac{1}{2} \mathbb{E}_{\boldsymbol{\Delta}_t}\Big[ \big( \frac{2}{1 + \mathrm{e}^{-\alpha_t \sqrt{\|\mathbf{g}_t\|^2 + \|\boldsymbol{\Delta}_t\|^2 + 2\mathbf{g}_t^\mathsf{T}\boldsymbol{\Delta}_t}}} - 1 \big) \frac{\mathbf{g}_t^\mathsf{T}\mathbf{g}_t + \mathbf{g}_t^\mathsf{T}\boldsymbol{\Delta}_t}{\sqrt{\|\mathbf{g}_t\|^2 + \|\boldsymbol{\Delta}_t\|^2 + 2\mathbf{g}_t^\mathsf{T}\boldsymbol{\Delta}_t}} \Big| \boldsymbol{\Delta}_t \in \mathcal{H}^+ \Big]$$

$$+ \frac{1}{2} \mathbb{E}_{\boldsymbol{\Delta}_t}\Big[ \big( \frac{2}{1 + \mathrm{e}^{-\alpha_t \sqrt{\|\mathbf{g}_t\|^2 + \|\boldsymbol{\Delta}_t\|^2 - 2\mathbf{g}_t^\mathsf{T}\boldsymbol{\Delta}_t}}} - 1 \big) \frac{\mathbf{g}_t^\mathsf{T}\mathbf{g}_t - \mathbf{g}_t^\mathsf{T}\boldsymbol{\Delta}_t}{\sqrt{\|\mathbf{g}_t\|^2 + \|\boldsymbol{\Delta}_t\|^2 - 2\mathbf{g}_t^\mathsf{T}\boldsymbol{\Delta}_t}} \Big| \boldsymbol{\Delta}_t \in \mathcal{H}^+ \Big], \tag{50}$$

where (a) follows from conditioning the expectation w.r.t. $\boldsymbol{\Delta}_t$ on two halfspaces, and using the fact that under the symmetric distribution assumption on $\boldsymbol{\Delta}_t$ (i.e., $\boldsymbol{\Delta}_t \overset{\mathcal{D}}{=} -\boldsymbol{\Delta}_t$ in Assumption 7.4), we have

$$\Pr\{\boldsymbol{\Delta}_t \in \mathcal{H}^+\} = \Pr\{\boldsymbol{\Delta}_t \in \mathcal{H}^-\} = \frac{1}{2}, \tag{51}$$

and (b) follows again from symmetric distribution assumption on $\boldsymbol{\Delta}_t$ and along with the variable exchange $\boldsymbol{\Delta}_t = -\boldsymbol{\Delta}_t$ in the second term of (49).

We now rewrite (50) in terms of the random variables $s$ and $c \in [-1, 1]$ introduced in the lemma. Noting that the event $\boldsymbol{\Delta}_t \in \mathcal{H}^+$ corresponds to $c \geq 0$, we can write

$$\frac{1}{2} \mathbb{E}_{\boldsymbol{\Delta}_t}\Big[ \big( \frac{2}{1 + \mathrm{e}^{-\alpha_t \sqrt{\|\mathbf{g}_t\|^2 + \|\boldsymbol{\Delta}_t\|^2 + 2\mathbf{g}_t^\mathsf{T}\boldsymbol{\Delta}_t}}} - 1 \big) \frac{\mathbf{g}_t^\mathsf{T}\mathbf{g}_t + \mathbf{g}_t^\mathsf{T}\boldsymbol{\Delta}_t}{\sqrt{\|\mathbf{g}_t\|^2 + \|\boldsymbol{\Delta}_t\|^2 + 2\mathbf{g}_t^\mathsf{T}\boldsymbol{\Delta}_t}} \Big| \boldsymbol{\Delta}_t \in \mathcal{H}^+ \Big]$$

$$+ \frac{1}{2} \mathbb{E}_{\boldsymbol{\Delta}_t}\Big[ \big( \frac{2}{1 + \mathrm{e}^{-\alpha_t \sqrt{\|\mathbf{g}_t\|^2 + \|\boldsymbol{\Delta}_t\|^2 - 2\mathbf{g}_t^\mathsf{T}\boldsymbol{\Delta}_t}}} - 1 \big) \frac{\mathbf{g}_t^\mathsf{T}\mathbf{g}_t - \mathbf{g}_t^\mathsf{T}\boldsymbol{\Delta}_t}{\sqrt{\|\mathbf{g}_t\|^2 + \|\boldsymbol{\Delta}_t\|^2 - 2\mathbf{g}_t^\mathsf{T}\boldsymbol{\Delta}_t}} \Big| \boldsymbol{\Delta}_t \in \mathcal{H}^+ \Big] \tag{52}$$

$$= \frac{\|\mathbf{g}_t\|}{2} \mathbb{E}_{s,c}\Big[ \frac{1 + sc}{\sqrt{1 + s^2 + 2sc}} \big( \frac{2}{1 + \mathrm{e}^{-\gamma_t \sqrt{1 + s^2 + 2sc}}} - 1 \big) \Big| 0 \leq c \leq 1 \Big]$$

$$+ \frac{\|\mathbf{g}_t\|}{2} \mathbb{E}_{s,c}\Big[ \frac{1 - sc}{\sqrt{1 + s^2 - 2sc}} \big( \frac{2}{1 + \mathrm{e}^{-\gamma_t \sqrt{1 + s^2 - 2sc}}} - 1 \big) \Big| 0 \leq c \leq 1 \Big], \tag{53}$$

where the equality follows directly from the definitions of $s$ and $c$. Expressing (53) in terms of $f(s, c, \gamma_t)$ yields the result. $\square$

**Lemma E.3.** *The function* $\frac{\psi_\alpha(x)}{x}$ *with* $\psi_\alpha(x)$ *defined in* (5) *is a non-increasing function in* $x$ *for* $\alpha \geq 0$, $x \geq 0$.

*Proof.* Computing the derivative of the function w.r.t. $x$ and simplifying the terms yields the following:

$$\frac{\partial(\psi_\alpha(x)/x)}{\partial x} = \frac{\partial \psi_\alpha(x)}{\partial x} \frac{1}{x} - \frac{1}{x^2} \psi_\alpha(x)$$

$$= \frac{2\mathrm{e}^{-\alpha x}(\alpha x - \sinh \alpha x)}{x^2 (1 + \mathrm{e}^{\alpha x})^2}, \tag{54}$$

where the RHS is a non-positive term, since $\alpha x \le \sinh \alpha x$ for $\alpha \ge 0$, $x \ge 0$. Thus, the derivative is non-positive, which shows that the function is non-increasing. $\qquad \square$

**Lemma E.4.** *The function $\frac{\partial \psi_\alpha(x)}{\partial x}$ with $\psi_\alpha(x)$ defined in (5) is a non-increasing function in $x$ for $\alpha \ge 0$, $x \ge 0$.*

*Proof.* We compute the derivative of the function as

$$
\begin{aligned}
\frac{\partial(\partial \psi_\alpha(x)/\partial x)}{\partial x} &= \partial\Big(\frac{2\alpha \mathrm{e}^{-\alpha x}}{\left(1 + \mathrm{e}^{-\alpha x}\right)^2}\Big)/\partial x \\
&= \frac{-2\alpha^2 \mathrm{e}^{-\alpha x}\left(1 - \mathrm{e}^{-\alpha x}\right)}{\left(1 + \mathrm{e}^{-\alpha x}\right)^3},
\end{aligned}
\tag{55}
$$

which is non-positive for $\alpha \ge 0$ and $x \ge 0$, which leads to the desired conclusion. $\qquad \square$

**Lemma E.5.** *For the function $\psi_\alpha(x)$ defined in (5), the following holds:*

$$
\frac{\partial \psi_\alpha(x)}{\partial x} \le \frac{\psi_\alpha(x)}{x}, \ \alpha \ge 0, x \ge 0.
\tag{56}
$$

*Proof.* We begin with the inequality $\alpha x \le \sinh \alpha x$ for $x \ge 0$ and $\alpha \ge 0$. We have

$$
\begin{aligned}
\alpha x \le \sinh \alpha x &\overset{\text{(a)}}{\Rightarrow} 2\alpha x \mathrm{e}^{-\alpha x} \le 1 - \mathrm{e}^{-2\alpha x} \\
&\overset{\text{(b)}}{\Rightarrow} \frac{2\alpha \mathrm{e}^{-\alpha x}}{\left(1 + \mathrm{e}^{-\alpha x}\right)^2} \le \frac{1}{x}\Big(\frac{2}{1 + \mathrm{e}^{-\alpha x}} - 1\Big) \\
&\overset{\text{(c)}}{\Rightarrow} \frac{\partial \psi_\alpha(x)}{\partial x} \le \frac{\psi_\alpha(x)}{x},
\end{aligned}
\tag{57}
$$

where (a) follows from multiplying both sides of the previous inequality by $2\mathrm{e}^{-\alpha x}$, (b) follows from dividing both sides by $x\left(1 + \mathrm{e}^{-\alpha x}\right)^2$, and (c) results from writing both sides in terms of $\psi_\alpha(x)$ and $\frac{\partial \psi_\alpha(x)}{\partial x}$. $\qquad \square$

**Theorem E.6.** *Let $f(s, c, \gamma_t)$ be the function defined in Lemma E.2. Then, the following properties hold:*

1. *$f(s, c, \gamma_t)$ is non-increasing in $s$ for $0 \le c \le 1$ and $\gamma_t \ge 0$.*

2. *$f(s, c, \gamma_t) \ge 0$ for $0 \le c \le 1$ and $\gamma_t \ge 0$.*

3. *$f(s, c, \gamma_t)$ is non-increasing in $c$ on the interval $0 \le c \le 1$, for $s \ge 1$ and $\gamma_t \ge 0$.*

*Proof.* We first prove the first property, showing that $f(s, c, \gamma_t)$ is non-increasing in $s$. Based on (44) in Lemma E.2, we rewrite $f(s, c, \gamma_t)$ as

$$
f(s, c, \gamma_t) = f_1(s, c)\psi_{\gamma_t}(x_1) + f_2(s, c)\psi_{\gamma_t}(x_2),
\tag{58}
$$

where we define $x_1 \triangleq \sqrt{1 + s^2 + 2sc}$, $x_2 \triangleq \sqrt{1 + s^2 - 2sc}$, and

$$
f_1(s, c) \triangleq \frac{1 + sc}{\sqrt{1 + s^2 + 2sc}},
\tag{59}
$$

$$
f_2(s, c) \triangleq \frac{1 - sc}{\sqrt{1 + s^2 - 2sc}}.
\tag{60}
$$

Taking the derivative of $f(s, c, \gamma_t)$ w.r.t. $s$, we have

$$
\begin{aligned}
\frac{\partial f(s, c, \gamma_t)}{\partial s} &= \frac{\partial f_1(s, c)}{\partial s}\psi_{\gamma_t}(x_1) + f_1(s, c)\frac{\partial \psi_{\gamma_t}(x_1)}{\partial x_1}\frac{\partial x_1}{\partial s} \\
&\quad + \frac{\partial f_2(s, c)}{\partial s}\psi_{\gamma_t}(x_2) + f_2(s, c)\frac{\partial \psi_{\gamma_t}(x_2)}{\partial x_2}\frac{\partial x_2}{\partial s}.
\end{aligned}
\tag{61}
$$

Substituting the derivatives into the above expression results in

$$\frac{\partial f(s,c,\gamma_t)}{\partial s} = \frac{-s(1-c^2)}{(1+s^2+2sc)^{\frac{3}{2}}}\psi_{\gamma_t}(x_1) + \frac{\partial \psi_{\gamma_t}(x_1)}{\partial x_1}\left(c + \frac{s(1-c^2)}{1+s^2+2sc}\right)$$
$$+ \frac{-s(1-c^2)}{(1+s^2-2sc)^{\frac{3}{2}}}\psi_{\gamma_t}(x_2) + \frac{\partial \psi_{\gamma_t}(x_2)}{\partial x_2}\left(-c + \frac{s(1-c^2)}{1+s^2-2sc}\right). \quad (62)$$

Rearranging the terms, we have

$$\frac{\partial f(s,c,\gamma_t)}{\partial s} = -c\left(\frac{\partial \psi_{\gamma_t}(x_2)}{\partial x_2} - \frac{\partial \psi_{\gamma_t}(x_1)}{\partial x_1}\right)$$
$$- \frac{s(1-c^2)}{1+s^2+2sc}\left(\frac{\psi_{\gamma_t}(x_1)}{x_1} - \frac{\partial \psi_{\gamma_t}(x_1)}{\partial x_1}\right)$$
$$- \frac{s(1-c^2)}{1+s^2-2sc}\left(\frac{\psi_{\gamma_t}(x_2)}{x_2} - \frac{\partial \psi_{\gamma_t}(x_2)}{\partial x_2}\right). \quad (63)$$

The first term in the RHS of (63) is non-positive since, by Lemma E.4, the derivative of $\psi_\alpha(x)$ is a non-increasing function of $x$, and we have $x_2 \leq x_1$ for $c \geq 0, s \geq 0$. The second and third terms are also non-positive because $c \leq 1$ and, by Lemma E.5, it holds that

$$\frac{\psi_{\gamma_t}(x)}{x} \geq \frac{\partial \psi_{\gamma_t}(x)}{\partial x}, \quad \forall x \geq 0, \forall \gamma_t \geq 0. \quad (64)$$

Thus, we conclude that $\frac{\partial f(s,c,\gamma_t)}{\partial s} \leq 0$, implying that $f(s,c,\gamma_t)$ is non-increasing in $s$.

To prove the second property, we first evaluate the limit of the function as $s$ approaches infinity:

$$\lim_{s\to\infty} f(s,c,\gamma_t) = 0, \ 1 \geq c \geq 0, \gamma_t \geq 0. \quad (65)$$

By the first property, $f(s,c,\gamma_t)$ is non-increasing in $s$, which implies that its minimum value is zero.

To establish the third property, we compute the derivative of $f(s,c,\gamma_t)$ w.r.t. $c$. Based on (58), we have

$$\frac{\partial f(s,c,\gamma_t)}{\partial c} = \frac{\partial f_1(s,c)}{\partial c}\psi_{\gamma_t}(x_1) + f_1(s,c)\frac{\partial \psi_{\gamma_t}(x_1)}{\partial x_1}\frac{\partial x_1}{\partial c} + \frac{\partial f_2(s,c)}{\partial c}\psi_{\gamma_t}(x_2)$$
$$+ f_2(s,c)\frac{\partial \psi_{\gamma_t}(x_2)}{\partial x_2}\frac{\partial x_2}{\partial c}. \quad (66)$$

Plugging the derivatives into the above expression and rearranging the terms, we obtain

$$\frac{\partial f(s,c,\gamma_t)}{\partial c} = \frac{\psi_{\gamma_t}(x_1)}{x_1}\frac{s^2(s+c)}{1+s^2+2sc} - \frac{\psi_{\gamma_t}(x_2)}{x_2}\frac{s^2(s-c)}{1+s^2-2sc}$$
$$+ \frac{\partial \psi_{\gamma_t}(x_1)}{\partial x_1}\frac{s(1+sc)}{1+s^2+2sc} - \frac{\partial \psi_{\gamma_t}(x_2)}{\partial x_2}\frac{s(1-sc)}{1+s^2-2sc}. \quad (67)$$

Next, we find an upper bound for $\frac{\partial f(s,c,\gamma_t)}{\partial c}$. Based on Lemma E.3, $\frac{\psi_{\gamma_t}(x)}{x}$ is a non-increasing function in $x$. Thus, we have

$$\frac{\psi_{\gamma_t}(x_1)}{x_1} \leq \frac{\psi_{\gamma_t}(x_2)}{x_2}, \quad (68)$$

which follows from the fact that $x_2 \leq x_1$ for $s \geq 0$ and $1 \geq c \geq 0$. Furthermore, by Lemma E.4, the function $\frac{\partial \psi_{\gamma_t}(x)}{\partial x}$ is non-increasing in $x$. Thus, since $x_2 \leq x_1$, it follows that

$$\frac{\partial \psi_{\gamma_t}(x_1)}{\partial x_1} \leq \frac{\partial \psi_{\gamma_t}(x_2)}{\partial x_2}. \quad (69)$$

Substituting the upper bounds from (68) and (69) into (67), we obtain

$$\frac{\partial f(s,c,\gamma_t)}{\partial c}$$
$$\leq \frac{\psi_{\gamma_t}(x_2)}{x_2}\left(\frac{s^2(s+c)}{1+s^2+2sc} - \frac{s^2(s-c)}{1+s^2-2sc}\right) + \frac{\partial \psi_{\gamma_t}(x_2)}{\partial x_2}\left(\frac{s(1+sc)}{1+s^2+2sc} - \frac{s(1-sc)}{1+s^2-2sc}\right)$$
$$= \frac{\psi_{\gamma_t}(x_2)}{x_2}\frac{2cs^2(1-s^2)}{(1+s^2-2sc)(1+s^2+2sc)} - \frac{\partial \psi_{\gamma_t}(x_2)}{\partial x_2}\frac{2cs^2(1-s^2)}{(1+s^2-2sc)(1+s^2+2sc)}. \quad (70)$$

The RHS of the upper bound in (70) is non-positive for $s \geq 1$ and $1 \geq c \geq 0$, since $\frac{\partial \psi_{\gamma_t}(x_2)}{\partial x_2} \leq \frac{\psi_{\gamma_t}(x_2)}{x_2}$ by Lemma E.5. $\qquad \square$

**Lemma E.7.** *For any $r \geq \|\mathbf{g}_t\|$, the following inequality holds:*

$$\mathbb{E}_{s,c}\Big[f(s,c,\gamma_t)\Big|0 \leq c \leq 1\Big] \geq f\Big(\frac{r}{\|\mathbf{g}_t\|}, c = 1, \gamma_t\Big) \Pr\Big\{s \leq \frac{r}{\|\mathbf{g}_t\|}\Big\}. \tag{71}$$

*Proof.* For any $r \in \mathbb{R}$, we have

$$\mathbb{E}_{s,c}\Big[f(s,c,\gamma_t)\Big|0 \leq c \leq 1\Big] \overset{(a)}{\geq} \mathbb{E}_{s,c}\Big[f(s,c,\gamma_t)\Big|0 \leq c \leq 1, s \leq \frac{r}{\|\mathbf{g}_t\|}\Big] \Pr\Big\{s \leq \frac{r}{\|\mathbf{g}_t\|}\Big\}$$

$$+ \mathbb{E}_{s,c}\Big[f(s,c,\gamma_t)\Big|0 \leq c \leq 1, s \geq \frac{r}{\|\mathbf{g}_t\|}\Big] \Pr\Big\{s \geq \frac{r}{\|\mathbf{g}_t\|}\Big\} \tag{72}$$

$$\overset{(b)}{\geq} \mathbb{E}_{s,c}\Big[f(s,c,\gamma_t)\Big|0 \leq c \leq 1, s \leq \frac{r}{\|\mathbf{g}_t\|}\Big] \Pr\Big\{s \leq \frac{r}{\|\mathbf{g}_t\|}\Big\} \tag{73}$$

$$\overset{(c)}{\geq} \mathbb{E}_c\Big[f\Big(\frac{r}{\|\mathbf{g}_t\|}, c, \gamma_t\Big)\Big|0 \leq c \leq 1\Big] \Pr\Big\{s \leq \frac{r}{\|\mathbf{g}_t\|}\Big\}, \tag{74}$$

where (a) follows from conditioning the expectation on the events $s \geq \frac{r}{\|\mathbf{g}_t\|}$ and $s \leq \frac{r}{\|\mathbf{g}_t\|}$, (b) is concluded by dropping the second term in (72) and noting that $f(s,c,\gamma_t) \geq 0$ based on Theorem E.6, and (c) follows from the fact that $f(s,c,\gamma_t)$ is non-increasing in $s$ for any $1 \geq c \geq 0$ and $\gamma_t \geq 0$ due to Theorem E.6.

We now restrict $r \geq \|\mathbf{g}_t\|$ and use the fact from Theorem E.6 that $f(s,c,\gamma_t)$ is non-increasing in $c$ for any $s \geq 1$ to conclude that for any $r \geq \|\mathbf{g}_t\|$,

$$\mathbb{E}_c\Big[f\Big(\frac{r}{\|\mathbf{g}_t\|}, c, \gamma_t\Big)\Big|0 \leq c \leq 1\Big] \Pr\Big\{s \leq \frac{r}{\|\mathbf{g}_t\|}\Big\} \geq f\Big(\frac{r}{\|\mathbf{g}_t\|}, c = 1, \gamma_t\Big) \Pr\Big\{s \leq \frac{r}{\|\mathbf{g}_t\|}\Big\}, \tag{75}$$

which completes the proof. $\qquad \square$

**Lemma E.8.** *For $s \geq 1$, $f(s, c = 1, \gamma_t)$ is lower bounded as*

$$f(s, c = 1, \gamma_t) \geq \frac{\gamma_t}{\cosh(s\gamma_t)}. \tag{76}$$

*Proof.* By substituting $c = 1$ into the definition of $f(s, c, \gamma_t)$ given in Lemma E.2 and simplifying the expression, we obtain

$$f(s, c = 1, \gamma_t)$$

$$= \frac{1+s}{\sqrt{1+s^2+2s}}\Big(\frac{2}{1+\mathrm{e}^{-\gamma_t\sqrt{1+s^2+2s}}} - 1\Big) + \frac{1-s}{\sqrt{1+s^2-2s}}\Big(\frac{2}{1+\mathrm{e}^{-\gamma_t\sqrt{1+s^2-2s}}} - 1\Big) \tag{77}$$

$$= \frac{2}{1+\mathrm{e}^{-\gamma_t(s+1)}} - \frac{2}{1+\mathrm{e}^{-\gamma_t(s-1)}} \tag{78}$$

$$= \frac{2\sinh(\gamma_t)}{\cosh(\gamma_t) + \cosh(s\gamma_t)}. \tag{79}$$

We next note that $s \geq 1$. Using the fact that $\cosh(x)$ is an increasing function on $x \geq 0$, we can conclude that $\cosh(s\gamma_t) \geq \cosh(\gamma_t)$. We then use the lower bound $\sinh(x) \geq x$ for $x \geq 0$ to write

$$\frac{2\sinh(\gamma_t)}{\cosh(\gamma_t) + \cosh(s\gamma_t)} \geq \frac{\gamma_t}{\cosh(s\gamma_t)}, \tag{80}$$

which completes the proof. $\qquad \square$

**Lemma E.9.** *Let $\lambda_\alpha = k_1/T$ and $\alpha_0 = k_2/r$ for some $k_1 > 0$ and $k_2 > 0$. Under Algorithm 1, the saturation slope is bounded on both sides as*

$$\frac{\kappa_1}{r} \leq \alpha_t \leq \frac{\kappa_2}{r}, \ 0 \leq t \leq T-1, \tag{81}$$

*where $\kappa_1 \triangleq k_2\mathrm{e}^{-k_1}$ and $\kappa_2 \triangleq k_2\mathrm{e}^{k_1}$.*

*Proof.* First, observe that for $t = 0$ we have $\alpha_0 = \frac{k_2}{r}$, which satisfies

$$\frac{\kappa_1}{r} \leq \alpha_0 \leq \frac{\kappa_2}{r},$$

since $k_1 > 0$. We next show that the same bound also holds for $1 \leq t \leq T-1$. Substituting $\lambda_\alpha = \frac{k_1}{T}$ and $\alpha_0 = \frac{k_2}{r}$ into the update rule (16), we obtain for any $t \geq 1$,

$$\alpha_t = \frac{k_2}{r} e^{\frac{k_1 q_t}{T}}, \tag{82}$$

where $q_t$ is defined as

$$q_t \triangleq \sum_{\tau=0}^{t-1} \text{sign}\big(\hat{s}(\boldsymbol{\theta}_\tau, \alpha_\tau; \mathcal{B}_\tau)^\mathsf{T} \hat{r}(\boldsymbol{\theta}_{\tau-1}, \alpha_{\tau-1}; \mathcal{B}_{\tau-1})\big), \tag{83}$$

with $\hat{r}(\boldsymbol{\theta}_{-1}, \alpha_{-1}; \mathcal{B}_{-1}) = \mathbf{0}$. Since $-t \leq q_t \leq t$ and $1 \leq t \leq T-1$, we have

$$\alpha_t \geq \frac{k_2}{r} e^{\frac{-k_1(T-1)}{T}} \geq \frac{k_2}{r} e^{-k_1} = \frac{\kappa_1}{r} \tag{84}$$

$$\alpha_t \leq \frac{k_2}{r} e^{\frac{k_1(T-1)}{T}} \leq \frac{k_2}{r} e^{k_1} = \frac{\kappa_2}{r}, \tag{85}$$

which completes the proof. $\qquad\square$

**Theorem E.10.** *Let Assumptions 7.1–7.4 hold, and suppose that $\lambda_\alpha \propto 1/T$ and $\alpha_0 \propto 1/r$ for some constant $r \geq G$. Then, under the DP-SGD-AdaSig algorithm described in Algorithm 1, the following inequality holds:*

$$\frac{1}{T} \sum_{t=0}^{T-1} \Pr\big\{\|\boldsymbol{\Delta}_t\| \leq r\big\} \mathbb{E}\|\mathbf{g}_t\|^2 \leq r\tilde{G}\Big(\frac{L(\boldsymbol{\theta}_0) - L^\star}{\lambda T} + \beta\lambda\big(1 + \frac{d\sigma_s^2}{2B^2}\big)\Big), \tag{86}$$

*where $\tilde{G} > 0$ is a constant.*

*Proof.* Using the results of Lemmas E.1, E.2, E.7, E.8, and E.9, we have

$$\frac{1}{\lambda}\mathbb{E}\Big[L(\boldsymbol{\theta}_t) - L(\boldsymbol{\theta}_{t+1})\Big|\boldsymbol{\theta}_t, \alpha_t\Big] + \beta\lambda\Big(1 + \frac{d\sigma_s^2}{2B^2}\Big)$$

$$\overset{\text{Lemma E.1}}{\geq} \mathbf{g}_t^\mathsf{T}\mathbb{E}\Big[\big(\frac{2}{1 + e^{-\alpha_t\|\mathbf{v}_t\|}} - 1\big)\frac{\mathbf{v}_t}{\|\mathbf{v}_t\|}\Big|\boldsymbol{\theta}_t, \alpha_t\Big] \tag{87}$$

$$\overset{\text{Lemma E.2}}{=} \frac{\|\mathbf{g}_t\|}{2}\mathbb{E}_{s,c}\Big[f(s, c, \gamma_t)\Big|0 \leq c \leq 1\Big] \tag{88}$$

$$\overset{\text{(a)}}{\geq} \frac{\|\mathbf{g}_t\|}{2}f\Big(\frac{r}{\|\mathbf{g}_t\|}, c = 1, \gamma_t\Big)\Pr\big\{\|\boldsymbol{\Delta}_t\| \leq r\big\} \tag{89}$$

$$\overset{\text{(b)}}{\geq} \frac{\alpha_t\|\mathbf{g}_t\|^2}{2\cosh(r\alpha_t)}\Pr\big\{\|\boldsymbol{\Delta}_t\| \leq r\big\} \tag{90}$$

$$\overset{\text{Lemma E.9}}{\geq} \frac{\kappa_1\|\mathbf{g}_t\|^2}{2r\cosh(\kappa_2)}\Pr\big\{\|\boldsymbol{\Delta}_t\| \leq r\big\}, \tag{91}$$

where (a) is obtained by applying Lemma E.7, which requires the bound $r \geq \|\mathbf{g}_t\|, \forall t$. This bound follows directly from Assumption 7.3 (i.e., $\|\mathbf{g}_t\| \leq G, \forall t$) together with the assumption $r \geq G$. Inequality (b) is obtained by applying Lemma E.8, using the bound $r \geq \|\mathbf{g}_t\|, \forall t$. The last inequality follows from Lemma E.9, since under the assumptions $\lambda_\alpha \propto 1/T$ and $\alpha_0 \propto 1/r$, we can equivalently write $\lambda_\alpha = k_1/T$ and $\alpha_0 = k_2/r$ for some constants $k_1 > 0$ and $k_2 > 0$. Substituting these expressions into the lemma, the inequality holds with $\kappa_1 = k_2 e^{-k_1}, \kappa_2 = k_2 e^{k_1}$.

Let us now define $\tilde{G} \triangleq 2\cosh(\kappa_2)/\kappa_1$. After taking expectation from both sides of the last inequality, summing over iterations from 0 to $T-1$, and dividing by $T$, we have

$$\frac{1}{r\tilde{G}T} \sum_{t=0}^{T-1} \Pr\Big\{\|\boldsymbol{\Delta}_t\| \leq r\Big\} \mathbb{E}\|\mathbf{g}_t\|^2$$

$$\leq \frac{1}{\lambda T} \sum_{t=0}^{T-1} \mathbb{E}\Big[L(\boldsymbol{\theta}_t) - L(\boldsymbol{\theta}_{t+1})\Big] + \frac{1}{T} \sum_{t=0}^{T-1} \beta\lambda\Big(1 + \frac{d\sigma_s^2}{2B^2}\Big) \tag{92}$$

$$\overset{(a)}{\leq} \frac{L(\boldsymbol{\theta}_0) - L^\star}{\lambda T} + \beta\lambda\Big(1 + \frac{d\sigma_s^2}{2B^2}\Big), \tag{93}$$

where (a) follows from Assumption 7.1. Rearranging the terms yields the inequality in (86). $\square$

## E.2 Proof of Theorem 7.5

*Proof.* We use the result in Theorem E.10 and set the noise multiplier $\sigma_s$ such that privacy is guaranteed. To ensure the privacy guarantee, we use the result in Theorem 1 in [1]. For clarity, we first restate Theorem 1 in [1].

**Theorem E.11** (Theorem 1 of [1]). *There exist constants $u$ and $\nu$ such that, given the sampling probability $q = B/N$, for any $\epsilon \leq uq^2T$, the composition of $T$ Gaussian mechanisms, each with noise multiplier $\sigma$, satisfies $(\epsilon, \delta)$-DP if*

$$\sigma^2 \geq \frac{\nu^2 q^2 T \log(1/\delta)}{\epsilon^2}. \tag{94}$$

According to Theorem E.11, the composition of $T$ Gaussian mechanisms ensures $(\epsilon, \delta)$-DP provided that the noise multiplier $\sigma$ is set to

$$\sigma^2 = \frac{\nu^2 B^2 T \log(1/\delta)}{N^2 \epsilon^2}. \tag{95}$$

Note that our proposed algorithm (DP-SGD-AdaSig, described in Algorithm 1) requires two noise multipliers, $\sigma_s$ and $\sigma_r$, since each iteration involves two Gaussian mechanisms, $\hat{s}(\cdot)$ and $\hat{r}(\cdot)$. According to Proposition 6.1, these two parallel Gaussian mechanisms are equivalent to a single Gaussian mechanism with noise multiplier $\sigma$, where

$$\frac{1}{\sigma^2} = \frac{1}{\sigma_s^2} + \frac{1}{\sigma_r^2}. \tag{96}$$

To ensure $(\epsilon, \delta)$-DP for the overall algorithm after $T$ iterations, we set $\sigma_s$ and $\sigma_r$ such that the equivalent noise multiplier $\sigma$ in (96) satisfies (95). This can be achieved by setting $\sigma_s$ and $\sigma_r$ as follows:

$$\sigma_s^2 = \frac{\nu_s^2 B^2 T \log(1/\delta)}{N^2 \epsilon^2}, \tag{97}$$

$$\sigma_r^2 = \frac{\nu_r^2 B^2 T \log(1/\delta)}{N^2 \epsilon^2}, \tag{98}$$

with constants $\nu_s$ and $\nu_r$ satisfying

$$\frac{1}{\nu^2} = \frac{1}{\nu_s^2} + \frac{1}{\nu_r^2}. \tag{99}$$

The choices of $\sigma_s$ and $\sigma_r$ in (97) and (98), together with constants $\nu_s$ and $\nu_r$ that satisfy the equality in (99), ensure that (96) holds and thereby guarantee $(\epsilon, \delta)$-DP for the DP-SGD-AdaSig algorithm.

Substituting $\sigma_s^2$ from (97) into (86) in Theorem E.10, we then optimize its LHS w.r.t. the learning rate $\lambda$, which yields

$$\lambda = \sqrt{\frac{2N^2\epsilon^2 (L(\boldsymbol{\theta}_0) - L^\star)}{\beta T \left(2N^2\epsilon^2 + d\nu_s^2 T \log\left(\frac{1}{\delta}\right)\right)}}. \tag{100}$$

Substituting this learning rate into the LHS concludes the result. $\square$

### E.3  Proof of Corollary 7.7

*Proof.* The probability $\Pr\left\{\|\boldsymbol{\Delta}_t\| \leq r\right\}$ can be lower bounded as

$$\Pr\left\{\|\boldsymbol{\Delta}_t\| \leq r\right\} \overset{(a)}{\geq} 1 - \frac{\mathbb{E}\|\boldsymbol{\Delta}_t\|}{r} \tag{101}$$

$$\overset{(b)}{\geq} 1 - \frac{\sqrt{\mathbb{E}\|\boldsymbol{\Delta}_t\|^2}}{r} \tag{102}$$

$$\overset{(c)}{\geq} 1 - \frac{\zeta}{r}, \tag{103}$$

where (a) follows from Markov's inequality, (b) follows from Jensen's inequality, and (c) is due to the bounded variance assumption.

Using the derived probability lower bound in (103) on the LHS of (18) in Theorem 7.5, and considering $r \geq \max\{\zeta, G\}$, we divide both sides by $1 - \frac{\zeta}{r}$ to obtain:

$$\frac{1}{T}\sum_{t=0}^{T-1}\mathbb{E}\|\mathbf{g}_t\|^2 \leq \frac{2\tilde{G}r^2}{r-\zeta}\sqrt{\frac{\beta\big(L(\boldsymbol{\theta}_0) - L^\star\big)}{T}\left(1 + \frac{d\nu_s^2 T\log(\frac{1}{\delta})}{2N^2\epsilon^2}\right)}. \tag{104}$$

We next set $r$ such that the RHS of the above inequality is minimized. Note that when $r \geq \zeta$, the function $\frac{r^2}{r-\zeta}$ achieves its minimum value at $r = 2\zeta$, and for $r \geq 2\zeta$, the function is increasing. Noting that in Theorem 7.5 $r \geq G$, two cases may occur: namely, $G \leq 2\zeta$ or $G \geq 2\zeta$. By considering these two cases, separately, we can conclude that

$$r_\star = \operatorname*{argmin}_{r \geq \max\{\zeta, G\}} \frac{r^2}{r-\zeta} = \begin{cases} 2\zeta, & \text{if } G \leq 2\zeta, \\ G, & \text{if } G \geq 2\zeta \end{cases} = \max\{2\zeta, G\}. \tag{105}$$

Substituting $r_\star$ into (104) completes the proof. $\qquad\square$

## F  Experiments Settings

This section provides further information on the experimental settings considered in the paper. We present more details on the datasets and architectures used for the numerical experiments presented in the paper.

### F.1  General Settings

**Privacy Settings**    In all experiments, we fix the DP budget $(\epsilon, \delta)$, and compute the noise multiplier, i.e., $\sigma$, numerically using the Opcaus library [40], such that the DP budget spent after $T$ iterations (or equivalently $Tq$ epochs) equals to the fixed budget $(\epsilon, \delta)$. Considering Proposition 6.1, for a given $\sigma$, different pairs of $\sigma_s$ and $\sigma_r$ can be obtained, where decreasing one increases the other. To simplify the parameter space, we set $\sigma_s = 1.01\sigma$ in all AdaSig experiments, making $\sigma_s$ smaller than $\sigma_r$. This choice prioritizes a more accurate update of the model over the saturation slope, as roughly updating $\alpha$ in the descent direction is sufficient.

**Hyperparameters**    To ensure fair comparison, we use the same batch size and number of epochs across all methods. For AdaSig, we tune the learning rates $\lambda$, $\lambda_\alpha$, and the initial saturation slope $\alpha_0$. For vanilla clipping, we tune $C$ and $\lambda$. For Auto-S and PSAC, we use the tuned value of $C$ for vanilla clipping following their original papers [4, 36], and tune hyperparameters $r$ and learning rate $\lambda$. For the method in [3], we tune the initial clipping threshold $C^0$, clipping threshold learning rate $\eta_C$, learning rate $\lambda$, $\sigma_b$, and $\gamma$.

### F.2  Settings for Image Classification

**MNIST and FashionMNIST**    We use the 4-layer CNN with $\tanh$ activation proposed in [27] and described in Table 6 of [34], with cross-entropy loss and the DP-SGD optimizer. The DP budget is set to $(\epsilon, \delta) = (3, 10^{-5})$. For batch size, number of epochs, and momentum, which are common across different methods, we use the values reported in [34]. These values are summarized in Table 2. Table 3 presents the best values of other hyperparameters for each method.

Table 2: Common hyperparameters across different methods used for training CNN on MNIST, FashionMNIST, and CIFAR-10.

| Parameter | MNIST | FashionMNIST | CIFAR-10 |
|---|---|---|---|
| Batch size ($B$) | 512 | 2048 | 4096 |
| Number of epochs ($Tq$) | 40 | 40 | 60 |
| Momentum | 0.9 | 0.9 | 0.9 |

Table 3: Hyperparameters selected for each method for training CNN on MNIST, FashionMNIST, and CIFAR-10.

| Method | Parameter | MNIST | FashionMNIST | CIFAR-10 |
|---|---|---|---|---|
| Vanilla | $C$ | 0.1 | 0.1 | 0.1 |
| | $\lambda$ | 0.5 | 4.0 | 3.0 |
| Auto-S | $r$ | 0.01 | 0.01 | 0.01 |
| | $C$ | 0.1 | 0.1 | 0.1 |
| | $\lambda$ | 0.5 | 4.0 | 3.0 |
| PSAC | $r$ | 0.1 | 0.1 | 0.1 |
| | $C$ | 0.1 | 0.1 | 0.1 |
| | $\lambda$ | 0.5 | 4.0 | 3.0 |
| Method in [3] | $C^0$ | 0.1 | 0.1 | 0.1 |
| | $\lambda$ | 0.5 | 4.0 | 3.0 |
| | $\eta_C$ | 0.05 | 0.01 | 0.01 |
| | $\sigma_b$ | 40.0 | 30.0 | 25.0 |
| | $\gamma$ | 0.5 | 0.5 | 1.0 |
| AdaSig | $\alpha_0$ | 5.0 | 1.0 | 1.0 |
| | $\lambda$ | 0.05 | 0.4 | 0.25 |
| | $\lambda_\alpha$ | 0.01 | 0.01 | 0.01 |

**CIFAR-10** We use the 8-layer CNN with tanh activation from [27], as detailed in Table 7 of [34], with cross-entropy loss and DP-SGD optimizer. The DP budget is set to $(\epsilon, \delta) = (3, 10^{-5})$. We use the same batch size, number of epochs, and momentum across different methods as reported in Table 2.[4] The best values of other hyperparameters for different methods are reported in Table 3.

**ImageNette** We use ImageNette, a 10-class subset of ImageNet [9], with an image size of $160 \times 160$. We consider ResNet-9 architecture (about 2.5 million parameters), with the Mish activation function [25] and cross-entropy loss. For training, the DP-Nesterov-accelerated Adam (DP-NAdam) optimizer is utilized.[5] We consider a DP budget of $(\epsilon, \delta) = (8, 10^{-4})$. We follow [17] by using group normalization instead of batch normalization without scale normalization. The only difference is that we do not apply the learning rate decay schedule. The ResNet-9 architecture can be found at [17].[6] All methods use the same batch size and number of epochs, which are set according to the values reported in [4, 36], as shown in Table 4. Table 5 lists the best values of the remaining hyperparameters for each method.

**CelebA** We use the same ResNet-9 architecture as for the ImageNette dataset, with the DP-Adam optimizer. The CelebA dataset contains 40 labels per image and we use it for both single-label and multi-label classification tasks.

---

[4]Note that the batch size and the number of epochs differ from those reported in [34] and used in [4, 36]. Our experiments showed that these hyperparameter changes yield better performance across all baselines compared with those reported in the references.

[5]For ImageNette and CelebA, we use more advanced optimizers than DP-SGD, namely, DP-NAdam and DP-Adam, respectively, to achieve improved performance for these more challenging datasets. These experiments further demonstrate the wide applicability of the AdaSig approach with different optimizers.

[6]Check also https://gist.github.com/gkaissis/ 6db6b7271f93d3459263b6978cfd4146.

Table 4: Common hyperparameters across different methods used for training Resnet-9 on ImageNette and CelebA.

| Parameter | ImageNette | CelebA (Multi-label/Smiling) |
|---|---|---|
| Batch size ($B$) | 1024 | 512 |
| Number of epochs ($Tq$) | 50 | 10 |

Table 5: Hyperparameters selected for each method for training ResNet-9 on ImageNette, and CelebA.

| Method | Parameter | ImageNette | CelebA(Multi-label) | CelebA (Smiling) |
|---|---|---|---|---|
| Vanilla | $C$ | 1.5 | 0.1 | 0.1 |
| | $\lambda$ | $5 \times 10^{-4}$ | $10^{-3}$ | $10^{-3}$ |
| Auto-S | $r$ | 0.01 | 0.01 | 0.01 |
| | $C$ | 1.5 | 0.1 | 0.1 |
| | $\lambda$ | $5 \times 10^{-4}$ | $10^{-3}$ | $10^{-3}$ |
| PSAC | $r$ | 0.1 | 0.1 | 0.1 |
| | $C$ | 1.5 | 0.1 | 0.1 |
| | $\lambda$ | $5 \times 10^{-4}$ | $10^{-3}$ | $10^{-3}$ |
| Method in [3] | $C^0$ | 1.5 | 0.1 | 0.1 |
| | $\lambda$ | $5 \times 10^{-4}$ | $5 \times 10^{-4}$ | $2 \times 10^{-3}$ |
| | $\eta_C$ | 0.01 | 0.01 | 0.05 |
| | $\sigma_b$ | 30.0 | 20.0 | 25.0 |
| | $\gamma$ | 3.0 | 1.0 | 1.0 |
| AdaSig | $\alpha_0$ | 5.0 | 1.0 | 1.0 |
| | $\lambda$ | $10^{-3}$ | $5 \times 10^{-4}$ | $5 \times 10^{-4}$ |
| | $\lambda_\alpha$ | 0.02 | 0.001 | 0.02 |

- For the single-label task, we perform binary classification considering the label `'Smiling'`, where we use the binary cross-entropy loss for training. In this scenario, the output layer of ResNet-9 consists of a single neuron.

- For the multi-label classification task, all available 40 labels are considered for prediction. Here, the output layer contains 40 neurons, and we use a scalar loss function that averages the 40 binary cross-entropy losses from each label.

In both cases, we set the DP budget to $(\epsilon, \delta) = (8, 5 \times 10^{-6})$.

The batch size and the number of epochs are set the same across all methods, following the settings in [4, 36], and are reported in Table 4. The best values of other hyperparameters for each method are presented in Table 5.

## F.3 Setting for Sentence Classification

**SST-2 and QNLI** We use a pre-trained RoBERTa-base model (about 125 million parameters) [22] and perform full parameter fine-tuning. We use the cross-entropy loss function. The AdamW optimizer [24] without weight decay is applied. A learning rate scheduler is used to linearly reduce the learning rate from its initial value to zero throughout the training process. The DP budget is set to $(\epsilon, \delta) = (3, \frac{1}{2N})$, where $N$ is the number of training samples for each dataset. It is worth mentioning that SST-2 and QNLI contain 67,349 and 104,743 training samples, respectively.

The batch size, number of epochs, and maximum sequence length for all methods are set according to [21, 4], and are given in Table 6. Table 7 lists the best value of other hyperparameters for each method.

Table 6: Common hyperparameters across different methods used for RoBERTa-base full parameter fine-tuning on SST-2 and QNLI.

| Parameter | SST-2 | QNLI |
|---|---|---|
| Batch size ($B$) | 1000 | 2000 |
| Number of epochs ($Tq$) | 3 | 6 |
| Max sequence length | 256 | 256 |

Table 7: Hyperparameters selected for each method for fine-tuning RoBERTAa-base on SST-2 and QNLI.

| Method | Parameter | SST-2 | QNLI |
|---|---|---|---|
| Vanilla | $C$ | 0.1 | 0.1 |
| | $\lambda$ | $5 \times 10^{-4}$ | $5 \times 10^{-4}$ |
| Auto-S | $r$ | 0.01 | 0.01 |
| | $C$ | 0.1 | 0.1 |
| | $\lambda$ | $5 \times 10^{-4}$ | $5 \times 10^{-4}$ |
| PSAC | $r$ | 0.1 | 0.1 |
| | $C$ | 0.1 | 0.1 |
| | $\lambda$ | $5 \times 10^{-4}$ | $5 \times 10^{-4}$ |
| Method in [3] | $C^0$ | 0.1 | 0.1 |
| | $\lambda$ | $5 \times 10^{-4}$ | $5 \times 10^{-4}$ |
| | $\eta_C$ | 0.01 | 0.01 |
| | $\sigma_b$ | 35.00 | 25.0 |
| | $\gamma$ | 1.0 | 1.0 |
| AdaSig | $\alpha_0$ | 1.0 | 1.0 |
| | $\lambda$ | $5 \times 10^{-4}$ | $5 \times 10^{-4}$ |
| | $\lambda_\alpha$ | 0.005 | 0.01 |

### F.4 Hardware and Software Information

All experiments are performed on a server equipped with Intel Xeon E5-2683 v4 CPUs, NVIDIA V100 GPUs, and 251 GiB of memory. The operating system used is AlmaLinux 9.3, and the CUDA Toolkit version is 12.2. The implementation of all training procedures is based on PyTorch 2.3.0 and Opacus 1.4.1.

## G Numerical Analysis of Direction Deviation and Magnitude Deviation

In this section, we provide additional figures to further assess the direction deviation and magnitude deviation achieved by the proposed AdaSig clipping method and the baselines.

### G.1 Defining Metrics

**Cosine Similarity**  To evaluate the direction deviation, we compute the cosine similarity between the aggregation of the clipped gradients, i.e., $\tilde{\mathbf{g}}_t \triangleq \sum_{i \in \mathcal{B}_t} \tilde{\mathbf{g}}_{i,t}$, and the true batch gradient, i.e., $\mathbf{g}_t \triangleq \sum_{i \in \mathcal{B}_t} \mathbf{g}_{i,t}$, as $\cos \phi_t = \frac{\langle \tilde{\mathbf{g}}_t, \mathbf{g}_t \rangle}{\|\tilde{\mathbf{g}}_t\| \cdot \|\mathbf{g}_t\|}$, which measures the cosine of the angle between $\tilde{\mathbf{g}}_t$ and $\mathbf{g}_t$.

**SNR**  To characterize the magnitude deviation, we define the signal-to-noise-ratio (SNR) as a normalized magnitude of the aggregation of the clipped gradients. Let us denote by $\|\tilde{\mathbf{g}}_t\|$ the magnitude of the aggregation of the clipped gradients after clipping, i.e., $\|\tilde{\mathbf{g}}_t\| = \left\| \sum_{i \in \mathcal{B}_t} \tilde{\mathbf{g}}_{i,t} \right\|$. We define the SNR to be the ratio of this magnitude to the standard deviation of the added privacy noise denoted by $\sigma_n$, i.e., $\text{SNR} = \frac{\|\tilde{\mathbf{g}}_t\|}{\sigma_n}$. For AdaSig, $\sigma_n = \sigma_s$. For vanilla clipping, Auto-S, and PSAC,

$\sigma_n = C\sigma$. For the method in [3], $\sigma_n = C^t z_\Delta$. This metric can describe the magnitude deviation against various approaches with potentially different privacy noise variances: as the SNR decreases, the performance degradation caused by the privacy noise becomes more severe.

## G.2 Performance Comparison

**ImageNette**   Figure 4 shows the histogram of the SNR during training on the ImageNette dataset. The results depict that using AdaSig, the SNR is concentrated around larger values. We further recall the earlier observations reported in Figure 3 in the main paper. There, we observe that using AdaSig, direction deviation is reduced compared with the baselines. Considering these two results, i.e., Figure 3 and Figure 4, we conclude that using AdaSig, the deteriorating impact of clipping is reduced with respect to *both* direction deviation and magnitude deviation. This demonstrates that, for ImageNette, AdaSig provides highly effective clipping that preserves closeness to the true batch gradient.

**CIFAR-10**   We next present the results of similar experiments on CIFAR-10. Figures 5 and 6 show the histogram of cosine similarity and SNR while training the 8-layer CNN on CIFAR-10, respectively. As observed in these figures, the cosine similarity distribution of AdaSig is skewed toward higher values relative to the baselines. This means that AdaSig incurs less direction deviation. Nevertheless, the histogram of the SNR achieved by AdaSig is close to those of the baselines. Thus, for CIFAR-10, AdaSig clipping is still able to improve the trade-off between the direction deviation and the magnitude deviation.

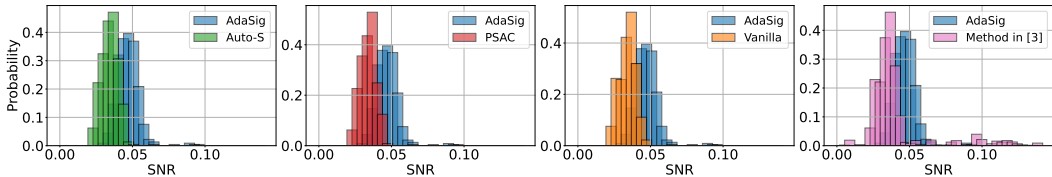

Figure 4: Comparison of normalized magnitude for different approaches during training ResNet-9 on ImageNette.

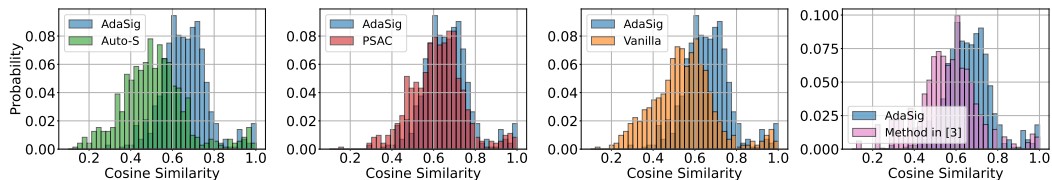

Figure 5: Comparison of cosine similarity for different approaches during training 8-layer CNN on CIFAR-10.

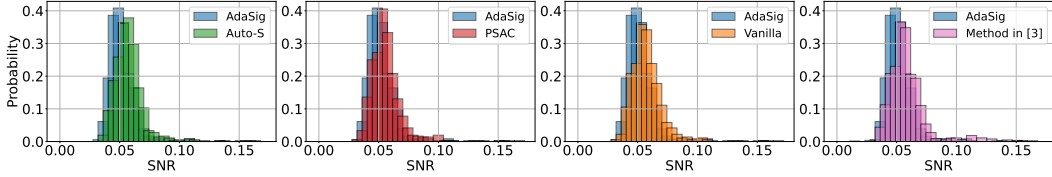

Figure 6: Comparison of normalized magnitude for different approaches during training 8-layer CNN on CIFAR-10.

# H Ablation Study: Sigmoid Clipping with Constant $\alpha$

An ablation method for the AdaSig approach is to use a fixed $\alpha$ across all iterations, i.e., $\alpha_t = \alpha, \forall t$. We examine sigmoid clipping with various fixed $\alpha$ values for training an 8-layer CNN on the CIFAR-10 dataset. The experimental setup is based on Appendix F.2.

## H.1 Impact of $\alpha$ on Direction Deviation and Magnitude Deviation

In this section, we study the impact of $\alpha$ on the *trade-off* between direction deviation and magnitude deviation. Figure 7 shows the histogram of cosine similarity and SNR during the training for three fixed values of $\alpha$. As seen, decreasing the $\alpha$ value results in an increase in cosine similarity, while reducing the SNR. This occurs because decreasing $\alpha$ expands the linear region of the sigmoid function, providing equal scaling for different gradient samples during clipping (see Figure 1 in the main paper), which reduces the direction deviation. This observation also aligns with the numerical example presented in Section 4 of the main paper.

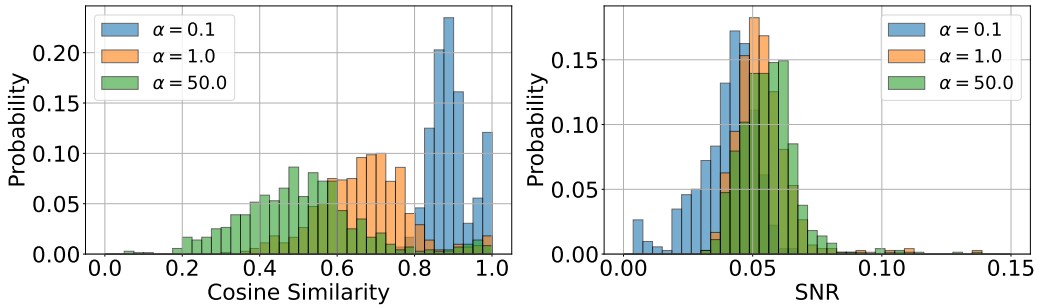

Figure 7: Trade-off between direction deviation and magnitude deviation for several fixed $\alpha$ values during the training of an 8-layer CNN on CIFAR-10. left: Cosine similarity histogram, right: SNR histogram.

## H.2 Impact of $\alpha$ on Test Accuracy

Next, we present the average final test accuracies for sigmoid clipping with several fixed $\alpha$ values in Table 8. As observed, very low values of $\alpha$ (less than 0.1) result in low accuracy due to a low SNR, while very large $\alpha$ values (greater than 5) also degrade accuracy due to direction deviation. However, for $\alpha$ in the middle range, we achieve the highest accuracy, resulting from a balance between direction deviation and magnitude deviation. It is worth noting that the highest accuracy achieved by $\alpha = 1$ is still lower than the accuracy attained by the AdaSig method, as shown in Table 1 in the main paper, which highlights that adaptively adjusting $\alpha$ over iterations further improves accuracy.

Table 8: Average test accuracies with 95% confidence intervals over five runs for sigmoid clipping with various fixed $\alpha$ values.

|  | $\alpha = 0.01$ | $\alpha = 0.1$ | $\alpha = 1.0$ | $\alpha = 5.0$ | $\alpha = 50.0$ | $\alpha = 100.0$ |
|---|---|---|---|---|---|---|
| Average test accuracy (%) | $45.27 \pm 0.67$ | $61.74 \pm 0.21$ | $62.13 \pm 0.12$ | $61.64 \pm 0.27$ | $60.62 \pm 0.38$ | $60.39 \pm 0.31$ |

# I Variation of $\alpha_t$ During Training

In this section, we illustrate how $\alpha_t$ evolves during training on CIFAR-10 for three different random seeds. As shown in Figure 8, the general trend across seeds is that $\alpha_t$ initially increases from its starting value 1.0 up to a peak in the middle of training, and then gradually decreases toward the end. The exact trajectory varies across seeds since each seed corresponds to a different batch sampling process, leading to distinct gradient statistics in each run.

When examining $\alpha_t$ over training iterations across different datasets and models, we do not observe a consistent pattern. This behavior arises from the dependence of $\alpha_t$ on individual sample gradients, which vary substantially across datasets and model architectures.

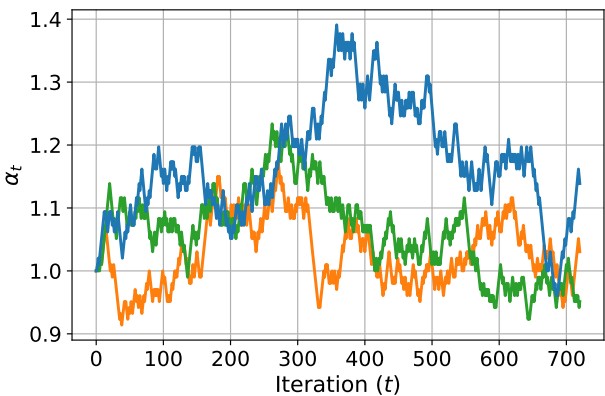

Figure 8: Evolution of $\alpha_t$ across training iterations on CIFAR-10 for three random seeds.

# J Convergence Plots

In this section, we present convergence plots for several experiments. Figure 9 illustrates the average test accuracy and test loss over five runs versus epochs during training on the CIFAR-10 dataset. The shaded regions around the curves represent the $95\%$ confidence intervals. As shown, the increasing test accuracy in the left plot and the decreasing test loss in the right plot demonstrate convergence for all methods, including AdaSig. Additionally, the figure highlights that AdaSig achieves a lower final test loss and higher final test accuracy compared with the baselines.

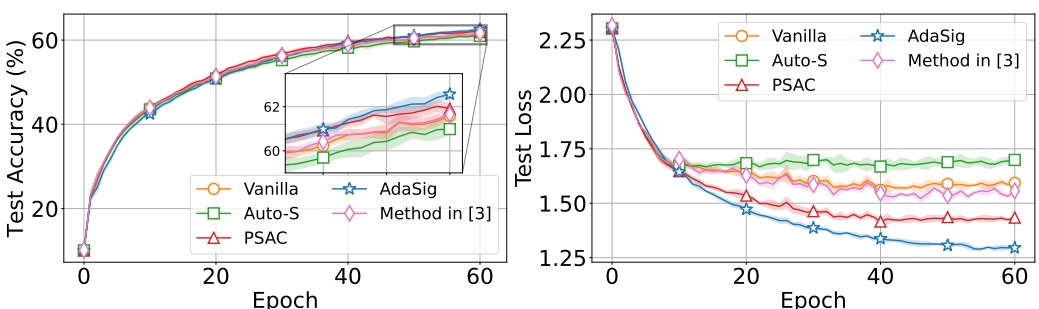

Figure 9: Convergence plot for CIFAR-10 dataset, left: Test accuracy vs. epoch, right: Test loss vs. epoch.

Additionally, Figure 10 shows the average test loss and accuracy throughout the training on the SST-2 dataset. As observed, all methods progressively reduce the test loss while increasing accuracy, with AdaSig outperforming the other methods.

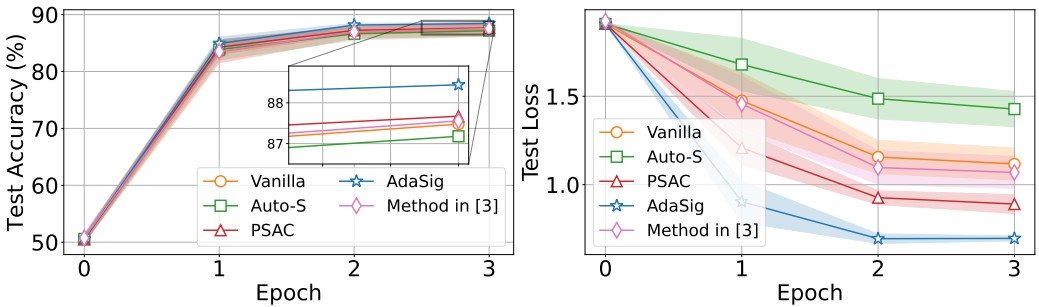

Figure 10: Convergence plot for SST-2 dataset, left: Test accuracy vs. epoch, right: Test loss vs. epoch.

