# OpenReview forum: "Adaptive Sigmoid Clipping for Balancing the Direction–Magnitude Mismatch Trade-off in Differentially Private Learning"
_NeurIPS.cc/2025/Conference — NeurIPS 2025 poster_

### Official Review · Reviewer_e8WR · 2025-06-28

**Clarity:** 3
**Significance:** 3
**Originality:** 3
**Rating:** 5
**Confidence:** 3

**Summary:**

The paper proposes a new adaptive clipping method for DP-SGD.

In the standard DP-SGD algorithm, a fixed clipping threshold is applied uniformly to each gradient in a batch.
This can lead to a significant mismatch between the direction of the sum of the clipped gradients and that of the original gradients, potentially resulting in degraded model performance.

Linear scaling offers a potential solution: if all gradients are scaled by the same factor, the direction of their sum is preserved.
This motivates the design of clipping mechanisms that approximate linear scaling for most gradients, while still enforcing an upper bound on gradients with large $\ell_2$ norms to ensure differential privacy after adding noise. Several prior works have explored this direction.

However, compared to fixed-threshold clipping, these methods often reduce the total $\ell_2$ norm of the clipped gradients more aggressively, which may in turn lower the signal-to-noise ratio once Gaussian noise is added.
Striking a balance between preserving the direction and maintaining the norm of the summed gradients remains a key challenge.

This paper addresses this challenge by introducing a new clipping mechanism based on the sigmoid function.
The method includes a parameter that explicitly controls the trade-off between preserving gradient direction and magnitude.
Moreover, this parameter is adaptively optimized for each batch in a differentially private manner.

Experimental results demonstrate that the proposed method outperforms existing state-of-the-art approaches across various benchmarks.

**Questions:**

Does the deep learning model studied in the experimental section (Section 8) satisfy the assumptions made in the theoretical analysis (Section 7)?

If not, it is unclear what the role of the theoretical section is, given that the motivation and intuition behind the proposed clipping method are already well-articulated earlier in the paper.
Clarifying the connection between the theory and the experiments would strengthen the overall contribution.

**Ethical Concerns:**

["NO or VERY MINOR ethics concerns only"]

**Final Justification:**

While I understand that the theoretical analysis aims to provide insight under standard assumptions, the paper would be strengthened by more clearly explaining how these assumptions hold or partially hold in practice.

I will keep my score unchanged.

**Limitations:**

See questions.

**Quality:**

3

**Strengths And Weaknesses:**

The paper studies an interesting problem and provides a concise and insightful description of existing solutions, clearly highlighting their advantages and limitations.
It is carefully written and easy to follow.

The proposed method for automatically updating the clipping threshold is elegant, and the experiments demonstrate improvements over the existing state-of-the-art.

The only shortcoming is that the theoretical section contributes little to the experimental results, and the connection between the two could be made stronger.

---

> ### Author Rebuttal · Authors · 2025-07-31
>
> We are grateful for the reviewer’s insights.
> > – Does the deep learning model studied in the experimental section (Section 8) satisfy the assumptions made in the theoretical analysis (Section 7)? If not, it is unclear what the role of the theoretical section is, given that the motivation and intuition behind the proposed clipping method are already well-articulated earlier in the paper. Clarifying the connection between the theory and the experiments would strengthen the overall contribution.
>
> The purpose of our theoretical analysis in Section 7 is to show that the proposed adaptive clipping retains the best-known convergence rate in the non-convex loss setting. This analysis serves as a basis for understanding the convergence behavior of the algorithm. It is natural to impose some assumptions on the loss function in order to theoretically characterize convergence. While these assumptions may not hold exactly in practice, they still offer valuable insight into how different parameters can affect the convergence bound.
>
> The assumptions we use—such as smoothness and bounded gradients—are standard in the literature [1-4] and broad enough to cover a wide range of models and loss functions. Although some of these may only partially hold for the deep learning models used in our experiments, the theoretical results remain informative and provide useful guidance for the design and behavior of our method in practical settings.
>
> We thank the reviewer for their suggestion on clarifying the role of the theoretical analysis, and we will reflect this discussion in the final version of the paper.
>
>
> [1] Zhiqi Bu, Yu-Xiang Wang, Sheng Zha, and George Karypis. Automatic clipping: Differentially private deep learning made easier and stronger. In NeurIPS, 2023.
>
> [2] Xiangyi Chen, Steven Z Wu, and Mingyi Hong. Understanding gradient clipping in private SGD: A geometric perspective. In NeurIPS, 2020.
>
> [3] Di Wang, MinweiYe, andJinhui Xu. Differentially private empirical risk minimization revisited: Faster and more general. In NeurIPS, 2017.
>
> [4] Xinwei Zhang, Zhiqi Bu, Steven Wu, and Mingyi Hong. Differentially private SGD without 440 441 clipping bias: An error-feedback approach. In ICLR, 2024.

---

> > ### Comment · Reviewer_e8WR · 2025-08-05
> >
> > Thank you for the clarification — while I understand that the theoretical analysis aims to provide insight under standard assumptions, the paper would be strengthened by more clearly explaining how these assumptions hold or partially hold in practice. I will keep my score unchanged.

---

### Official Review · Reviewer_8LLB · 2025-06-30

**Clarity:** 3
**Significance:** 2
**Originality:** 2
**Rating:** 4
**Confidence:** 3

**Summary:**

This work proposes AdaSig, an adaptive clipping strategy for Differentially Private Stochastic Gradient Descent (DP-SGD). Specifically, AdaSig clips the sampled gradients based on a sigmoid function for a proper balance between direction deviation and magnitude deviation. Extensive experiments on both sentence and image classification tasks demonstrate that AdaSig consistently outperforms existing clipping methods.

**Questions:**

Overall, I have the following questions, labeled as Q1-Q4. If all these concerns are addressed convincingly, I would increase my rating of this paper.

Q1. Is it necessary to adopt the sigmoid function in this context? What are the unique benefits of using it compared to other functions, such as $\tilde{\mathbf{g}}_{i,t} = \frac{C \mathbf{g}_{i,t}}{\Vert \mathbf{g}_{i,t} \Vert + r}$
        in [3] and
$\tilde{\mathbf{g}}_{i,t} = \frac{C \mathbf{g}_{i,t}}{\Vert \mathbf{g}_{i,t} \Vert + \frac{r}{\Vert \mathbf{g}_{i,t} \Vert + r}}$ in [32]. A theoretical comparison and experimental results would be helpful.

Q2. Would introducing a learnable parameter $r$ in the equation $\tilde{\mathbf{g}}_{i,t} = \frac{C \mathbf{g}_{i,t}}{\Vert \mathbf{g}_{i,t} \Vert + \frac{r}{\Vert \mathbf{g}_{i,t} \Vert + r}}$
         in [32] also achieve comparable performance to AdaSig? It would be valuable to include experimental results evaluating a learnable $r$ in [32] for a direct comparison.

Q3. What is the main technical difference between AdaSig and the method in [11]? AdaSig appears to be a simple application of the method in [11]. A more detailed explanation would be beneficial for better understanding the novelty of AdaSig.

Q4. How does $\alpha$ evolve throughout the training process? Providing a traceable trajectory of $\alpha$ could offer valuable insights into the adaptive behavior of the model.

**Ethical Concerns:**

["NO or VERY MINOR ethics concerns only"]

**Limitations:**

Yes.

**Paper Formatting Concerns:**

None.

**Quality:**

2

**Strengths And Weaknesses:**

Strengths (labeled as S1, S2, S3, etc.):

S1. This paper addresses a practical and significant problem, i.e., how to clip gradients during training to achieve a better trade-off between magnitude and direction deviations.

S2. The workflow of this paper is clearly presented, and the related works discussed in this paper are comprehensive and representative.

S3. The experiments are thorough, and the results provide compelling evidence for the effectiveness of the proposed method.

Weaknesses (labeled as W1, W2, W3, etc.):

W1. The authors propose a clipping operation based on the sigmoid function, and treat $\alpha$ as a learnable parameter for adjusting the deviations adaptively. However, it appears that the existing solution [32] can also reduce the direction deviation effectively. The key difference is that its linear region span remains fixed, as the corresponding parameter $r$ is not learnable. So is it necessary to leverage the sigmoid function? It seems that a learnable parameter $r$ in the equation
$$\tilde{\mathbf{g}}_{i,t} = \frac{C \mathbf{g}_{i,t}}{\Vert \mathbf{g}_{i,t} \Vert + \frac{r}{\Vert \mathbf{g}_{i,t} \Vert + r}}$$
     in [32] could also achieve comparable performance. A discussion on the necessity of the sigmoid function and the experiments on the learnable $r$ in [32] would be helpful.

W2. The authors focus on the trade-off between magnitude and direction deviations, but the proposed method appears to be an application or extension of that in [11]. What is the main technical difference between AdaSig and the method in [11]? A more detailed explanation would be beneficial for clarifying the novelty and contribution of the proposed method.


W3. The authors treat $\alpha$ as a learnable parameter for adjusting the deviations adaptively. Although Appendix H analyzes the impact of different $\alpha$ values, the evolution of $\alpha$ during the training process still remains unclear. Providing a traceable trajectory of $\alpha$ during training would bring valuable insights.

W4. There are a few writing issues in the paper.

* The title of Section 3 contains a typo: "Backgound" should be corrected to "Background".


* In line 127 in Section 3, a period is missing between "output" and "The".

* In Table 1, the values in AdaSig column should be consistently highlighted in bold.

---

> ### Author Rebuttal · Authors · 2025-07-31
>
> We appreciate the reviewer’s feedback.
>
> > Q1. Is it necessary to adopt the sigmoid function in this context? What are the unique benefits of using it compared to other functions, such as $\tilde{\mathbf{g}}{i,t} = \frac{C \mathbf{g}{i,t}}{\Vert \mathbf{g}{i,t} \Vert + r}$
> in [3] and $\tilde{\mathbf{g}}{i,t} = \frac{C \mathbf{g}_{i,t}}{\Vert \mathbf{g}{i,t} \Vert + \frac{r}{\Vert \mathbf{g}{i,t} \Vert + r}}$ in [32]. A theoretical comparison and experimental results would be helpful.
>
>
> The sigmoid function is simple yet well-suited to our goal of adaptively balancing the trade-off between direction and magnitude deviations. In particular, it enables control over the range of the linear span in Fig. 1 through the parameter $\alpha$. The sigmoid function provides greater flexibility and control compared with the clipping functions proposed in [32] and [3], even if those functions are extended with learnable parameters.
>
> The structure of the clipping function in [32] is such that the span of its linear region changes within a small range as $r$ varies. We are not allowed to include plots at this stage, but this effect is visible when the function is plotted for various values of $r$. We will incorporate these plots in the final version of the paper. This behavior can also be understood by analyzing the function in extreme cases. Due to the presence of the term $\frac{r}{x + r}$ in the denominator of PSAC, varying $r$ from $0$ to $\infty$ causes the function to transition between two limiting values: $1$ and $\frac{x}{x + 1}$. Hence, the span of its linear region is restricted between these two curves, and adaptively updating $r$ does not effectively balance the trade-off between direction and magnitude deviations. In contrast, the sigmoid function used in AdaSig can effectively span from values close to 1 down to curves near 0, thereby flexibly balancing the trade-off.
>
> While the clipping function in [3] does not suffer from this limited range of the linear span, it is less flexible than the sigmoid function in balancing the trade-off, as its values are less responsive to changes in $r$. Therefore, learning the optimal parameter requires greater variation across different values, whereas the sigmoid function, with its exponential change in $\alpha$, provides more flexible control over the linear region and thus, the direction-magnitude mismatch trade-off.
>
>
> > Q2. Would introducing a learnable parameter $r$ in the equation $\tilde{\mathbf{g}}{i,t} = \frac{C \mathbf{g}{i,t}}{\Vert \mathbf{g}{i,t} \Vert + \frac{r}{\Vert \mathbf{g}{i,t} \Vert + r}}$ in [32] also achieve comparable performance to AdaSig? It would be valuable to include experimental results evaluating a learnable $r$ in [32]  for a direct comparison.
>
> We first would like to clarify that deriving a differentially private update for the parameter of the clipping function is not straightforward and involves several steps that depend on the specific form of the clipping function. Specifically, the loss function must first be expressed in terms of the learnable parameter to compute its derivative w.r.t. that parameter. Then, to construct the private update, it is necessary to identify a constant upper bound on this derivative to guarantee bounded $\ell_2$-sensitivity, which allows the update to be privatized by adding a suitable amount of Gaussian noise.
>
> Following the reviewer’s suggestion, we have carried out the derivations and conducted experiments for the method in [32]; however, since the calculations exceed the allowed space, we are unable to include them here. Furthermore, due to the extensive tuning required to find the optimal initial value for $r$, we were only able to perform experiments on the CIFAR-10 dataset. The best accuracy obtained by this method was $61.96 \pm 0.15\%$, which is not significantly different from that of the original method. Considering the limited span of the linear region in [32], as discussed in our response to Q1, we believe these results are intuitive, since even with a learnable $r$, it has limited capacity to balance the direction-magnitude mismatch trade-off.
>
> We will conduct experiments on other datasets and include the results in the final version of the paper if the reviewer considers them necessary.
>
>
> > Q3. What is the main technical difference between AdaSig and the method in [11]? AdaSig appears to be a simple application of the method in [11]. A more detailed explanation would be beneficial for better understanding the novelty of AdaSig.
>
>
> First, we emphasize that AdaSig does **not** use the method in [11]. In [11], vanilla clipping is extended by optimizing the clipping threshold over iterations to balance the trade-off between bias and variance. Therefore, [11] does not introduce a new clipping methodology, but rather a new approach to applying vanilla clipping. Furthermore, it is unclear whether the method in [11] provides a theoretical convergence guarantee, as none is given in that paper.
>
> Despite both [11] and our work optimizing a learnable parameter, there are three key differences between them:  (i) AdaSig introduces a novel clipping methodology, whereas [11] builds on existing vanilla clipping as mentioned above.  (ii) The optimized parameters differ, with AdaSig tuning the saturation slope and [11] tuning the clipping threshold.  (iii) The purpose of the optimization is fundamentally different. AdaSig aims to balance the trade-off between the two components of bias, namely the direction and magnitude deviations, while keeping the variance fixed. In contrast, [11] seeks to balance the trade-off between bias and variance. More specifically, since AdaSig uses a sigmoid-based clipping function, optimizing the saturation slope $\alpha$ does not affect the output range. As a result, the $\ell_2$ sensitivity remains constant when changing $\alpha$ in AdaSig, leading to a fixed variance for the Gaussian noise used in the privacy mechanism. On the other hand, in [11], updating the clipping threshold changes the $\ell_2$ sensitivity, which in turn affects the variance of the added privacy noise.
>
>
>
>
>
> > Q4. How does $\alpha$ evolve throughout the training process? Providing a traceable trajectory of $\alpha$ could offer valuable insights into the adaptive behavior of the model.
>
> Since we are not allowed to include figures in our response, we provide a verbal description of the $\alpha_t$ behavior here and will include the plots in the final version.
>
> For example, on CIFAR-10, we observe that the average $\alpha_t$ increases from the initial value of 1.0 to just below 1.5 within the first 20 iterations. It then steadily decreases until around iteration 200, after which it begins to rise again, ultimately surpassing 1.1 by the end.
>
> Observing the changes in $\alpha_t$ over iterations across various datasets and models, we find no consistent trend in its variation. We conjecture that this is due to $\alpha_t$'s dependence on the gradients of individual samples, which can vary significantly across different datasets and model architectures.
>
>
> We thank the reviewer for pointing out the typos. We will carefully revise the manuscript to correct them.

---

> > ### Comment · Reviewer_8LLB · 2025-08-03
> >
> > The author's rebuttal address most of my concerns, so I will keep the positive rating score.

---

### Official Review · Reviewer_R2gd · 2025-07-03

**Clarity:** 2
**Significance:** 2
**Originality:** 3
**Rating:** 4
**Confidence:** 3

**Summary:**

The authors propose a learnable sigmoid‐based clipping scheme for DP-SGD that smoothly trades off gradient direction and magnitude errors. Upon privately learning the sigmoid’s slope $\alpha$ in addition to the model weights, they provide a sharp non-convex convergence rates while improving on the accuracy-privacy trade-off.

**Questions:**

1.) How are the seemingly similar looking performance numbers in Table 1 justified as an improvement? Please provide some additional context to help understand the suggested gains. For example PSAC 86.35 ± 0.19 Vs. proposed method at  86.68 ± 0.04 in the 2nd row of Table 1 with Fashion MNIST.

2.) How does the following clipping result for DP compare with your proposed result? https://arxiv.org/pdf/2502.11682 . Please cite and compare. Similarly, do so with https://arxiv.org/pdf/2505.20817? which is for a narrower class of functions.

**Ethical Concerns:**

["NO or VERY MINOR ethics concerns only"]

**Final Justification:**

Due to the effect of alpha being unclear in the experiments, I uphold my score. The authors have done a good rebuttal on rest of the points.

**Limitations:**

They haven't compared against this paper: https://arxiv.org/pdf/2502.11682 where authors claim that for non-convex smooth distributed problems with clients having arbitrarily heterogeneous data, they achieve an optimal convergence rate and also near optimal (local-)DP neighborhood.

**Paper Formatting Concerns:**

Figure 3 caption: ImageNette should be ImageNet.

**Quality:**

3

**Strengths And Weaknesses:**

Strength:
They provide the following equivalence in the extreme case showing an $O(\sqrt{d} /(N \varepsilon))$ that matched the vanilla DP-SGD in the asymptotic case when the iterations $T$ tend to infinity. The convergence analysis is done in good detail.

Room for improvement:
1.) The authors show that privacy accounting leads to the same effect in proposed method and DP-SGD when $\sigma_r=\left(\sigma^{-2}-\sigma_s^{-2}\right)^{-1 / 2}$. But they don't comment on the other settings, on how these two compare. For instance, the work in https://arxiv.org/pdf/2205.13710 shows that privacy-loss or degradation converges and is quite an ideal property to have as opposed to an unending loss.

Current privacy analyses of Noisy-SGD give an $(\alpha, \varepsilon)$-RDP upper bound of $$
\varepsilon \lesssim \frac{\alpha L^2}{n^2 \sigma^2} T$$ which increases ad infinitum as the number of iterations $T \rightarrow \infty$. The Altschuler-Talwar work above contains that increase as it shows a cut-off where the privacy loss begins to converge.

How does this above issue and corresponding sota result benefit from your adaptive clipping based version of DP-SGD?

2.) How does the following clipping result for DP compare with your proposed result? https://arxiv.org/pdf/2502.11682 . Please cite and compare. Similarly, do so with https://arxiv.org/pdf/2505.20817? which is for a narrower class of functions.

---

> ### Author Rebuttal · Authors · 2025-07-31
>
> We thank the reviewer for their valuable feedback.
> > Room for improvement 1.)  RDP bound in Noisy-SGD in [1]:
>
> We appreciate the reviewer’s reference to [1], which provides an important contribution by showing that the RDP privacy loss of Noisy-SGD can converge to a bounded value as the number of iterations $T \to \infty$, in contrast to standard analyses where the privacy loss grows unbounded.
>
> However, it is important to note that the analysis in [1] assumes **unclipped** gradients, as also mentioned in the discussion section of their paper. In contrast, our work focuses on the **clipped** setting; specifically, our analysis focuses on how adaptive clipping influences both training convergence and privacy guarantees.
>
> We believe extending the theoretical results for RDP analysis in [1] to the clipped regime remains an open and nontrivial direction. Once addressed, it could potentially open the door to analyzing more advanced clipping strategies, including our proposed AdaSig method.
>
>
> > Room for improvement 2.) How does the following clipping result for DP compare with your proposed result? [2]. Please cite and compare. Similarly, do so with [3]?  which is for a narrower class of functions.
>
> We thank the reviewer for pointing out these two recent works [2] and [3]. We will cite these two works to suggest potential directions for future research. Inspired by [2], one possible direction is to extend our AdaSig approach to the federated learning (FL) framework. Moreover, drawing on the methodology of [3], developing a high-probability convergence analysis would be a valuable theoretical extension of our work. Below, we highlight the key distinctions between our work and these papers:
>
>
> **Reference [2]:**
>
> The work in [2] considers a Federated Learning (FL) setting and proposes an algorithm that combines vanilla clipping, momentum, and error feedback. The authors show that this combination achieves an optimal convergence rate under client heterogeneity.
>
> While this is an important contribution for the FL setting, we note that the paper does *not* introduce a new clipping strategy, i.e., it relies on standard (vanilla) clipping. In contrast, our work focuses on the design of a new adaptive clipping mechanism, AdaSig, which dynamically balances direction and magnitude deviations in the clipped gradients, while maintaining optimal convergence guarantees. We note that a comparison with vanilla clipping used in [2] is already included in our experiments.
>
> Moreover, our work focuses on centralized (non-distributed) learning, whereas [2] addresses the FL setting, where client heterogeneity is a key challenge. This challenge does not arise in our scenario. That said, our proposed AdaSig clipping strategy is general and can also be applied in the FL setting. Like any clipping method, it can be combined with techniques such as error feedback and momentum to enhance performance under heterogeneous clients. We view this as a promising direction for future studies.
>
>
>
> **Reference [3]:**
>
> The work in [3] provides a high-probability convergence bound for vanilla clipped SGD under $(L_0, L_1)$-smoothness assumptions and heavy-tailed noise. In contrast, our work focuses on proposing a new adaptive clipping strategy, AdaSig, which aims to balance the distortion introduced by clipping in both gradient direction and magnitude, and establishes a convergence bound for this adaptive method.
>
> From a theoretical perspective, we note that the high-probability convergence analysis in [3] for vanilla clipping could potentially be extended to our adaptive clipping method. However, this extension is non-trivial and represents a promising direction for future work.
>
> From an experimental standpoint, we emphasize that our proposed method has already been compared against vanilla clipping, which is the clipping method in [3].
>
> >- How are the seemingly similar-looking performance numbers in Table 1 justified as an improvement? Please provide some additional context to help understand the suggested gains. For example, PSAC 86.35 ± 0.19 Vs. proposed method at 86.68 ± 0.04 in the 2nd row of Table 1 with Fashion MNIST.
>
> We would like to clarify that we performed a statistical $t$-test to assess the significance of the results. Specifically, the bold entries in Table 1 indicate cases where the $p$-value is less than 0.05, demonstrating that the observed improvements are statistically significant [4].
>
> In the specific example mentioned (second row), although the mean values are close (86.35 vs. 86.68), the $p$-value is below 0.05, confirming the statistical significance of the improvement. Furthermore, our method achieves a smaller confidence interval compared with PSAC in this case, indicating greater stability. We also note that in several other cases (e.g., rows 3, 4, and 7), the improvements over PSAC are more pronounced.
>
>
> > – Figure 3 caption: ImageNette should be ImageNet.
>
> We clarify that ImageNette [5] is a publicly available subset of ImageNet. It consists of 10 easily distinguishable classes from the original ImageNet dataset and is commonly used when full ImageNet training is computationally prohibitive. We will include a citation to [5] in the final version of the paper for clarity.
>
>
> [1] J. Altschuler and K. Talwar, “Privacy of noisy stochastic gradient descent: More iterations without more privacy loss,” Advances in Neural Information Processing Systems, vol. 35, pp. 3788–3800, 2022.
>
> [2] R. Islamov, S. Horvath, A. Lucchi, P. Richtarik, and E. Gorbunov, “Double Momentum and Error Feedback for Clipping with Fast Rates and Differential Privacy,” arXiv preprint arXiv:2502.11682, 2025.
>
> [3] S. Chezhegov, A. Beznosikov, S. Horváth, and E. Gorbunov, “Convergence of Clipped-SGD for Convex $(L_0, L_1)$-Smooth Optimization with Heavy-Tailed Noise,” arXiv preprint arXiv:2505.20817, 2025.
>
> [4] Greenland, S., et al. Statistical tests, P values, confidence intervals, and power: a guide to misinterpretations. In European journal of epidemiology, 2016.
>
> [5] J. Howard, ImageNette: A subset of 10 easily classified classes from ImageNet, fast.ai, 2019.

---

> > ### Comment · Reviewer_R2gd · 2025-08-04
> > **Thank you for comments. Further questions remain.**
> >
> > Thanks to authors for their comments. Upon closely checking through your proofs in the appendix, further questions arise:
> > 1.) In appendix C, around equations 20, 21, the approximation $\alpha_ {t−1} ​ ≈ \alpha_{t}$ ​ is invoked but not properly justified. Especially, in the initial steps when the convergence has not been achieved, while the gap remains when the chain-rule is invoked.
> > 2.) w.r.t equation 90, in appendix the proposition 6.1 requires $$
> > \frac{1}{\sigma_s^2}+\frac{1}{\sigma_r^2}=\frac{1}{\sigma^2}
> > $$ but the proof does not verify that $\sigma_s^2$ and $\sigma_r^2$ satisfy this for the chosen $\sigma$. This seems to be a slack in the privacy guarantee that has not been accounted for.
> >
> > I will reconsider my final scores based on the authors comments on this.

---

> > > ### Author Response · Authors · 2025-08-05
> > >
> > > We are grateful to the reviewer for further inspecting our paper. Below, we provide our answers to your questions.
> > >
> > >
> > > > Q1:
> > >
> > > We would like to clarify that this approximation is only used to motivate the derivation of the update rule for $\alpha$ in the proposed algorithm. This approximation is justified by the fact that, according to the update equation in (15), under a sufficiently small learning rate for $\alpha$ (denoted by $\lambda_\alpha$), the change in $\alpha_t$ between consecutive iterations is small, making the approximation $\alpha_{t-1} \approx \alpha_t$ reasonably accurate.
> > >
> > > We would like to emphasize that our privacy and convergence bound analyses do **not** rely on the exactness of this approximation. In particular, our analyses apply exactly to the proposed algorithm, whether or not $\alpha_{t-1} \approx \alpha_t$.
> > >
> > > We will add clarifications on this point in the revised version.
> > >
> > >
> > > > Q2:
> > >
> > > We would like to clarify that $\sigma_s$ and $\sigma_r$ are parameters of the proposed algorithm (Algorithm 1 in Appendix A) that are explicitly chosen to satisfy the relation $\frac{1}{\sigma_s^2} + \frac{1}{\sigma_r^2} = \frac{1}{\sigma^2}$, as stated in Proposition 6.1. Therefore, it is not necessary to verify whether this condition holds in the proof, as $\sigma_s$ and $\sigma_r$ are selected by us to ensure that this equality is satisfied.
> > >
> > >
> > > Specifically, for a given privacy budget $(\epsilon, \delta)$, we first compute the corresponding noise multiplier $\sigma$ for DP-SGD. Then, we set $\sigma_s$ and $\sigma_r$ in our algorithm (Algorithm 1 in Appendix A) such that they satisfy the above relation. This ensures that our algorithm satisfies $(\epsilon, \delta)$-DP.
> > >
> > > We will include clarifications on this point in the revised version, where the algorithm is defined.
> > >
> > >
> > >
> > > We hope we have answered your questions. Please let us know if any further clarification is required.

---

> > > > ### Comment · Reviewer_R2gd · 2025-08-05
> > > > **Thank you. I uphold my score**
> > > >
> > > > OK do include this clarification in the revised version (especially the second point with the different $\sigma$'s). Thank you for the information. I uphold my score thereby.

---

### Official Review · Reviewer_n68q · 2025-07-03

**Clarity:** 3
**Significance:** 2
**Originality:** 3
**Rating:** 4
**Confidence:** 3

**Summary:**

The paper addresses an important question in differentially private SGD. In standard DPSGD, each gradient is clipped to a fixed length C, to restrict the sensitivity of the corresponding data point.

However, this clipping means large gradients are clipped more than small ones, thereby fundamentally affecting the direction of the SGD path. The alternative is to scale down all gradients such that the largest gradients are at most C. But in this process, the smaller gradients become tiny, affecting the magnitude of the SGD steps.

The paper proposes a non-uniform scaling given by the sigmoid eq (5) and (6) that is meant to address the tradeoff of keeping sensitivity bounded, retaining some of the relative importance of large vs small gradients, and ensuring the small gradients are not too small.

The parameter $\alpha$ in the equation is updated dynamically during the training process in a privacy preserving manner.

The paper includes theorems about privacy guarantees and convergence of the training.

**Questions:**

1. Can you explain why a dynamic adaptation to $\alpha$ is needed and why a fixed $\alpha$ does not suffice?
2. Can you explain what your algorithm is to update $\alpha$?
3. Can you explain the novelty of the idea with respect to Auto-S, NSGD and PASC? These also seem to address the same issues with somewhat similar ideas. I read the discussion in related works, but did not quite catch the difference. Examples may help.

**Ethical Concerns:**

["NO or VERY MINOR ethics concerns only"]

**Final Justification:**

After the discussion, I feel my intitial understanding of the paper was accurate enough. I am thus retaining the rating and support acceptance of the paper.

I am have not checked the proofs in detail.

**Limitations:**

Yes.

**Quality:**

3

**Strengths And Weaknesses:**

Strengths

The paper is on an important and timely topic. It contributed interesting ideas to the area. It has theoretical guarantees for privacy and convergence. Experimental results on a range of models and data show good performance.

Weaknesses:

The need for adaptive changes to $\alpha$ are not explained. The process for deriving the right $\alpha$ in 5.2 is also unclear.

Minor comments:

The explanations in lines 68-75 and 152-166 could be made more clear. They are important for understanding the paper, but not easy to follow.

---

> ### Author Rebuttal · Authors · 2025-07-31
>
> We thank the reviewer for their comments and address their concerns below.
>
> > 1- Can you explain why a dynamic adaptation to $\alpha$  is needed and why a fixed $\alpha$ does not suffice?
>
> The value of $\alpha$ affects the span of the linear region, as illustrated in Fig. 1: a smaller $\alpha$ leads to a larger linear region (e.g., see the curves for $\alpha = 0.8$ and $\alpha = 10.0$). Thus, varying $\alpha$ allows us to adjust the span of the linear region and control the trade-off between direction and magnitude deviations throughout training. In contrast, a fixed value of $\alpha$ results in a static linear region span. Since the distribution of sample gradient norms $||\mathbf{g}_{i,t}||$ can vary significantly during training, using a fixed $\alpha$ across all iterations cannot effectively balance this trade-off at all times.
>
> To make this more concrete, consider Fig.1 and assume we have selected a fixed $\alpha = 0.8$ across all iterations. After several iterations, the maximum norm of the sample gradients within a batch may decrease significantly. In this case, it becomes possible to shrink the span of the linear region to reduce the magnitude deviation without causing excessive direction deviation. Recall from Fig. 1 that smaller linear regions, which correspond to larger $\alpha$ values such as $\alpha = 10$, introduce less magnitude deviation and more direction deviation. Thus, when the gradient norms are smaller, a larger $\alpha$ (that is, a narrower linear region) becomes preferable, as the risk of large direction deviation is mitigated due to the smaller gradient magnitudes. It is worth mentioning that this is just an illustrative case; in practice, gradient statistics may vary arbitrarily over time. Furthermore, the optimal adjustment of $\alpha$ depends not only on the maximum norm but on the overall distribution of the sample gradients. Our algorithm leverages this richer information to dynamically update $\alpha$ during training.
>
> We have experimentally validated the benefit of using an adaptive $\alpha$ in Appendix H.2 in the supplementary file. In Table 8, we show that the performance achieved with a fixed $\alpha$ remains consistently below that of our adaptive approach, highlighting the benefit of dynamically adjusting $\alpha$ during training.
>
>
>
> > 2- Can you explain what your algorithm is to update $\alpha$?
>
>
> Our algorithm updates $\alpha$ to reduce the learning loss while preserving privacy. In brief, the procedure for updating $\alpha$ in the $t$-th iteration is as follows:
>
> 1. Computing $\hat{s}_t$ which is defined in equation (9b). This value is already computed in the model updating step in equation (10).
> 2. Computing $\alpha_{t+1}$ from equation (15), using the value of $\hat{s_t}$ and $\hat{r}_{t-1}$, which was computed in the previous step.
> 3. Computing $\hat{r}_t$ using equation (13) and the definition of $r$ provided in Lemma 5.1. $\hat{r}_t$ will be used in the next iteration for $\alpha$ update in step 2.
>
> For further clarity, we kindly refer the reviewer to Algorithm 1 in Appendix A. We will also include additional clarifications in the algorithm description in the final version of the paper to make this procedure more transparent.
>
>
>
> > 3- Can you explain the novelty of the idea w.r.t. Auto-S, NSGD and PASC? These also seem to address the same issues with somewhat similar ideas. I read the discussion in related works, but did not quite catch the difference. Examples may help.
>
>
> Auto-S clipping (also known as NSGD) was proposed to eliminate the need for tuning the clipping threshold $C$ in vanilla clipping, which affects the bias-variance trade-off. This clipping operation is **not** designed to mitigate the *deviation effects* introduced by clipping. As shown in Fig. 1, Auto-S has a small linear region, which leads to significant direction deviation. This observation has also been validated numerically in [1].
>
> Later, PSAC clipping was proposed to address the direction mismatch of Auto-S by assigning smaller weights to sample gradients with smaller norms. As shown in Fig. 1, the PSAC curve has a wider linear region to serve this purpose. However, the PSAC clipping has a **fixed** linear region; that is, it cannot adaptively expand or shrink its linear region to balance the trade-off between direction and magnitude deviations (see also our response to the first question).
>
> In contrast, our proposed clipping method is based on an **adaptive** sigmoid function that can dynamically balance the aforementioned trade-off by adjusting its linear span through varying $\alpha$.
>
>
> > Minor comments: The explanations in lines 68-75 and 152-166 could be made more clear. They are important for understanding the paper, but not easy to follow.
>
>
> In lines 68–75, we discuss the issue of magnitude deviation and its trade-off with the direction deviation. The span of the linear region in Fig. 1 governs this trade-off. A larger linear region—corresponding to smaller values of $\alpha$—reduces direction deviation but increases magnitude deviation. This occurs because, for gradients with small norms (points close to 0 on the horizontal axis), the red curve (with a larger linear region) scales down the norm of the clipped gradients more severely than the green curve (with a smaller linear region), as shown in Fig. 1. This effect is further illustrated in Fig. 2(c) and 2(d). The dashed green arrow in Fig. 2(c) corresponds to the aggregation of clipped gradients for a large $\alpha$ value (i.e., smaller linear region), the dashed red arrow in Fig. 2(d) represents the aggregation of clipped gradients with a small $\alpha$ value (i.e., larger linear region), and the dashed blue arrow shows the summation of unclipped gradients. As shown, the dashed green arrow has a closer norm to the dashed blue arrow compared with the dashed red arrow. In contrast, the dashed red arrow exhibits reduced direction mismatch relative to the dashed blue arrow, but at the cost of increased magnitude deviation.  In summary, adjusting the value of $\alpha$ controls the size of the linear region and thus allows for balancing the trade-off between direction deviation and magnitude deviation.
>
> Lines 152-166 also refer to the same aforementioned trade-off, providing further elaboration. We will revise the indicated sections of the paper to enhance clarity.
>
>
> [1] Tianyu Xia, Shuheng Shen, Su Yao, Xinyi Fu, Ke Xu, Xiaolong Xu, and Xing Fu. Differentially 426 private learning with per-sample adaptive clipping. In Proc. of the AAAI Conference on Artificial 427 Intelligence, 2023.

---

> ### Comment · Reviewer_n68q · 2025-08-04
>
> I have no further comments. I have to admit that I do not completely get some aspects, but there does not seem to be any major flaws.

---

### Comment · Area_Chair_4GzB · 2025-08-05

Dear Reviewers,

Thank you for your valuable reviews. With the Reviewer-Author Discussions deadline approaching, please take a moment to read the authors' rebuttal and the other reviewers' feedback, and participate in the discussions and respond to the authors. Finally, be sure to complete the "Final Justification" text box and update your "Rating" as needed. Your contribution is greatly appreciated. I will flag irresponsible (final) reviews and/or any reviewers not participating in discussions.

Reviewers are expected to stay engaged in discussions, initiate them and respond to authors’ rebuttal, ask questions and listen to answers to help clarify remaining issues.

It is not OK to stay quiet.

It is not OK to leave discussions till the last moment.

If authors have resolved your (rebuttal) questions, do tell them so.

If authors have not resolved your (rebuttal) questions, do tell them so too.

Thanks.

AC

---

### Note · Authors · 2025-08-14

We believe we have thoroughly addressed all the main concerns raised by the reviewers. We are grateful to them for their valuable feedback and constructive discussion.

---

### Decision · Program_Chairs · 2025-09-17

**Decision:**

Accept (poster)

**Comment:**

This work proposes AdaSig, an adaptive clipping strategy for Differentially Private Stochastic Gradient Descent (DP-SGD). Specifically, AdaSig clips the sampled gradients based on a sigmoid function for a proper balance between direction deviation and magnitude deviation. Extensive experiments on both sentence and image classification tasks demonstrate that AdaSig consistently outperforms existing clipping methods.

Strength: (1) The paper is on an important and timely topic. It contributed interesting ideas to the area. It has theoretical guarantees for privacy and convergence. Experimental results on a range of models and data show good performance.

(2)  The workflow of this paper is clearly presented, and the related works discussed in this paper are comprehensive and representative.

Weakness: (1)The theoretical section contributes little to the experimental results.

As the clipping operator is very important in DP-SGD, the paper makes a fundamental contribution to the DP community. Although there are some improvement of the paper such as a finer theoretical analysis or the scalability issues. I would like to accept the paper.